# Pathogenic variants in the cohesin loader subunit MAU2 underlie a distinct Cornelia de Lange Syndrome subtype

The role of the cohesin complex depends on the cohesin loader proteins NIPBL and MAU2. While *NIPBL* variants are a major cause of Cornelia de Lange Syndrome (CdLS), the role of *MAU2* in disease is unclear. We describe 18 individuals carrying 15 heterozygous *MAU2* variants and demonstrate pathogenicity through functional analyses. In-frame *MAU2* variants predominantly impair NIPBL–MAU2 interaction, whereas truncating variants cause *MAU2* haploinsufficiency and lead to NIPBL reduction. Most individuals exhibit a DNA methylation profile compatible with the CdLS episignature. We also describe two *MAU2*-specific episignatures that reflect variant-dependent molecular consequences. Affected individuals display a wide range of phenotypes, from classic CdLS to milder presentations, with short stature and microcephaly as major features. A heterozygous *Mau2* knockout mouse model recapitulates these traits, confirming the causal role of *MAU2* disruption in vivo. Our study establishes *MAU2* as a CdLS-associated gene and delineates a *MAU2*-related chromatinopathy with variable expressivity.

Cornelia de Lange Syndrome (CdLS, OMIM #122470, #300590, #610759, #614701, #300882, and #620568) is a rare multisystem neurodevelopmental disorder with an estimated incidence between 1:10,000 and 1:30,000 live births[1]. The syndrome is characterized by distinctive craniofacial features, along with developmental delay, intellectual disability, behavioral anomalies, growth restriction, microcephaly, limb anomalies, and hirsutism[1]. The number and severity of these features vary among individuals, underscoring the broad clinical variability characteristic of CdLS[1]. An international consensus statement has developed a scoring system to standardize the clinical assessment of the syndrome, thereby facilitating diagnosis and evaluation of disease severity[1].

CdLS is primarily associated with pathogenic variants in subunits or regulators of cohesin, an evolutionarily conserved complex essential for sister chromatid cohesion, DNA repair, chromatin architecture, and transcriptional regulation[2]. CdLS appears to result primarily from disruption of transcriptional regulation and chromatin organization rather than cohesion, as cells of affected individuals show global chromatin and transcriptional dysregulation without cohesion defects. This dysregulation provides the basis for classifying CdLS as a chromatinopathy[3–5].

Pathogenic variants in six genes are known to cause CdLS[1]. Among these, the cohesin loader subunit NIPBL (OMIM #608667) is the primary contributor, with disease-causing variants identified in approximately 70% of affected individuals. Pathogenic variants in *NIPBL* are often associated with classic manifestations of the disease[1]. Up to 15% of cases are instead attributed to pathogenic variants in the *SMC1A*, *SMC3*, *RAD21*, *HDAC8*, and *BRD4* genes (OMIM #300040, #606062, #606462, #300269, and #608749)[1,6]. Pathogenic variants in these genes are associated with a wide clinical expressivity, including mild presentations as well as non-classical phenotypes[1].

The primary CdLS-associated protein, NIPBL, forms a heterodimeric complex with MAU2 (OMIM #614560)[7]. MAU2 enwraps the N-terminal region of NIPBL, promoting its proper folding and enabling the assembly of a functional cohesin loader complex[8]. This interaction is crucial for the stability of both proteins and the depletion of either component ultimately results in the downregulation of the other[9]. Despite the close functional interdependence between NIPBL and MAU2, the frequency of pathogenic variants differs strikingly: while 683 *NIPBL* variants are reported in the Human Gene Mutation Database (HGMD)[10], only two have been described in *MAU2*. We have

✉ e-mail: ilaria.parenti@uk-essen.de

previously reported the first individual with a disease-causing in-frame deletion in *MAU2* and a classic CdLS phenotype[11]. More recently, a second individual carrying a missense substitution was described[12]. This individual presented with a CdLS score warranting molecular testing, but with a rather atypical clinical presentation[12]. In addition, a few microdeletions at 19p13.11-p12 have been identified in individuals with neurodevelopmental delay. Some of these deletions encompass *MAU2* along with additional genes, making it challenging to delineate the specific contribution of *MAU2* to the resulting phenotypes[13]. In view of the available data, the precise role of *MAU2* in the context of disease onset remains unclear.

In this study, we report 18 individuals with 15 heterozygous variants in *MAU2*. Through a combination of detailed phenotyping, episignature analysis, functional assays, and the generation of a mouse model, we characterize this cohort and assess the molecular and phenotypic consequences of *MAU2* variants. Our findings demonstrate that the majority of these variants give rise to a CdLS subtype mainly characterized by short stature and microcephaly. Based on these observations, we propose that *MAU2* should be recognized as a CdLS-associated gene.

## Results

### Short-reads high-throughput sequencing identifies *MAU2* variants

We report 15 distinct heterozygous variants in the *MAU2* gene across 18 individuals. gnomAD v4.1.0[14] constraint metrics predict *MAU2* intolerance to both loss-of-function (pLI = 1) and missense variation (Z-score = 4.86), consistent with the presence of both truncating and in-frame variants in our cohort. The collected variants comprise five missense substitutions (p.(Cys50Ser), p.(Leu381Pro), p.(Cys394Tyr), p.(Leu528Phe), and p.(Asp534His)), four in-frame deletions (p.(Val54_Pro56del), p.(Ala309_Lys322del), p.(Lys322del), and p.(Gln310_Ala316del)), one in-frame delins (p.(Ala378_Gln380delinsAsp)), one nonsense variant (p.(Gln538*)), two frameshift duplications (p.(Cys145Leufs*14) and p.(Asp242Glyfs*28)), and two frameshift deletions, one resulting in a premature stop codon (p.(Gln135Argfs*32)), and one in protein elongation (p.(Ala601Leufs*124)) (Fig. 1). Only the p.(Cys50Ser) variant was found in gnomAD v4.1.0 (6 alleles), indicating that the *MAU2* variants reported here are extremely rare in population databases. Notably, the individual carrying the p.(Gln310_Ala316del) in-frame deletion, already previously described by our group[11], is included in the present cohort as well (Individual 1) to allow a more comprehensive clinical and molecular characterization of the *MAU2* variants.

The residues affected by missense substitutions and in-frame deletions localize to regions intolerant to variation, as indicated by residue-level intolerance profiles from MetaDome[15] (Fig. 1). The potential impact of the missense variants was additionally assessed using six in silico prediction tools (Supplementary Data 1). Each variant was classified as deleterious by at least three of these tools.

The variants occurred de novo in eight individuals (Individuals 1, 3, 4, 5, 9, 10, 13, and 17) (Supplementary Data 2). In five cases, the inheritance remained undetermined due to the unavailability of one or both parents for genetic testing (Individuals 2, 7, 8, 12, and 18). The remaining five variants were inherited from a parent. For Individuals 11 and 16, sufficient clinical information was available for the affected carrier parents, who were therefore included in the cohort as Individuals 12 and 17, respectively. The lack of sufficient clinical data for the carrier father of Individuals 14 and 15, who are siblings, and of Individual 6, precluded the inclusion of these two parents in the cohort. Through application of the relevant ACMG criteria, variants were classified as pathogenic in four individuals, likely pathogenic in eleven, and of uncertain significance in three (Supplementary Data 2).

### *MAU2* variant carriers exhibit a CdLS episignature

Variants in four of the six primary CdLS-associated genes (*NIPBL*, *SMC1A*, *SMC3*, and *RAD21*) result in overlapping DNA methylation profiles, collectively referred to as the CdLS episignature[16]. This prompted us to investigate whether variants in *MAU2* might result in the same methylation pattern. To explore this, we analyzed blood-derived DNA from 13 individuals of our cohort, as well as from the carrier father of Individuals 14 and 15.

EpiSign analysis revealed that ten cases exhibited a genome-wide DNA methylation profile similar to CdLS cases (Fig. 2A–C). According to the EpiSign interpretation framework, a result is considered positive when at least two of the following criteria are met: (1) elevated methylation variant pathogenicity (MVP) score, (2) hierarchical clustering with positive controls, and (3) multidimensional scaling (MDS) proximity to reference cases. Although several samples had MVP scores below the typical threshold of 0.5 (Fig. 2D), they were still classified as positive for the CdLS episignature because the other two analysis parameters (hierarchical clustering and MDS) matched the disorder-specific positive controls. The lower MVP scores observed in these cases may reflect a milder or variant-specific methylation profile, in which epigenetic alterations are present but less pronounced across the full CdLS episignature. Alternatively, this may be attributed to the

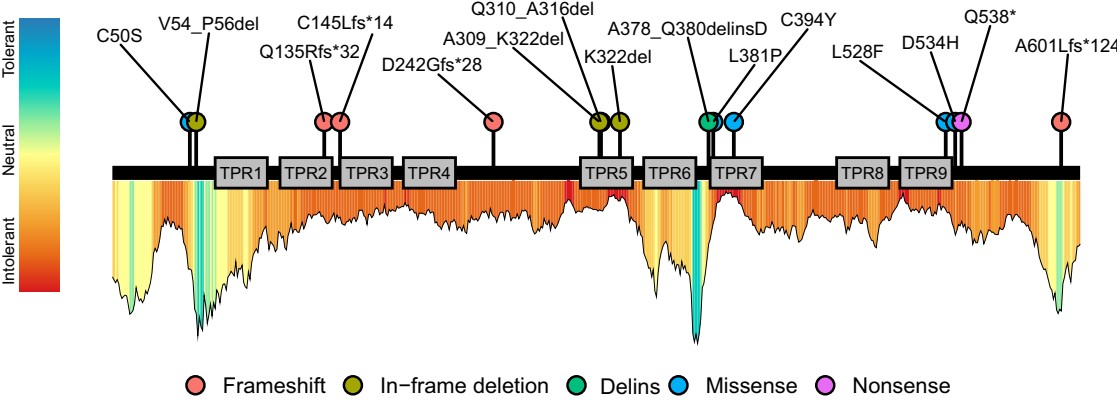

**Fig. 1 | Schematic representation of the MAU2 protein and its tetratricopeptide repeats (TPR), with the relative positions of the variants identified in this study.** Variant types are represented by colored lollipops: frameshift deletions/duplications (coral), in-frame deletions (olive green), delins (green), missense variants (blue), and nonsense variants (violet). A variation intolerance map derived from MetaDome[15] (https://stuart.radboudumc.nl/metadome/) is also presented beneath the protein structure. In this map, both the height and color of the curve indicate tolerance to missense variation, with valleys and red/orange colors representing intolerant regions, and peaks and green/blue colors representing tolerant regions.

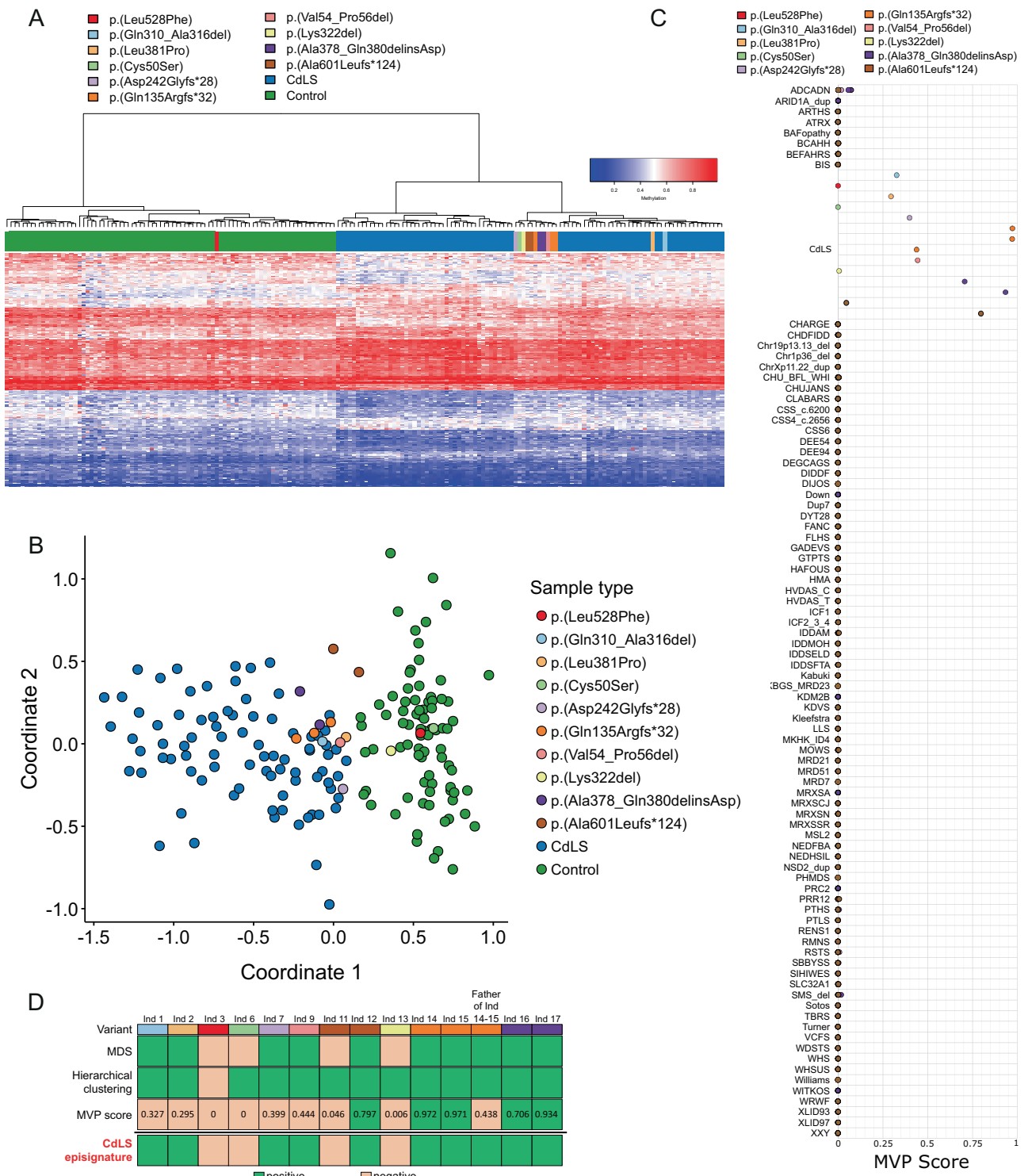

**Fig. 2 | EpiSign (DNA methylation) analysis of peripheral blood from 14 individuals with 10 unique variants in *MAU2*. A** Hierarchical clustering reveals that the *MAU2* variant carriers exhibit a DNA methylation profile that is more similar to subjects with a confirmed CdLS episignature (blue), than with controls (green). One case (red) does not cluster with the rest of the cohort. **B** Multidimensional scaling analysis demonstrates that 10 cases cluster with the confirmed CdLS cohort (blue) and are distinct from controls (green). Four cases negative for the CdLS episignature (red, light green, yellow, brown) cluster with controls (green). **C** 10 out of 14 cases display elevated MVP scores for CdLS. Four cases (red, light green, yellow, brown) show scores close to 0, suggesting a DNA methylation profile more similar to controls than to the CdLS cohort. **D** Summary of the DNA methylation analysis results with respect to the CdLS episignature. Colors match those assigned to each variant in Panel (**C**).

fact that the canonical CdLS episignature was not originally defined using cases with *MAU2* variants.

Four cases negative for the CdLS episignature (p.(Leu528Phe), p.(Cys50Ser), p.(Lys322del), and p.(Ala601Leufs*124)) exhibited

methylation profiles closer to controls, consistent with their lower MVP scores. Three of these variants (p.(Cys50Ser), p.(Lys322del), and p.(Ala601Leufs124)) clustered together with other *MAU2* variant carriers on the CdLS side of the heatmap (Fig. 2A) but did

not meet EpiSign criteria, as their MVP scores were below the positivity threshold (Fig. 2C) and they grouped with controls in the MDS analysis (Fig. 2B). To further assess their concordance with the CdLS episignature, the four *MAU2* samples classified as negative were visualized together against CdLS reference cases and controls, without inclusion of the other CdLS-positive *MAU2* samples (Supplementary Fig. 1). By this, all four samples grouped with controls in both the hierarchical clustering and the MDS plot, highlighting that these unsupervised visualization methods cluster samples based on their relative similarity to all other samples included in the analysis.

Since three of the four samples that did not align with the CdLS episignature clustered with CdLS cases when other *MAU2* samples were included in the hierarchical clustering analysis, we wondered whether there might be a *MAU2*-specific episignature. To address this, an episignature discovery analysis using *MAU2*-specific probes was performed. An initial discovery analysis including all 14 available samples did not yield clear separation from controls, suggesting heterogeneity in methylation changes across the full *MAU2* cohort. We therefore pursued an iterative discovery strategy aimed at identifying a subset of *MAU2* samples with sufficiently consistent methylation differences to define a reproducible episignature. This revealed two *MAU2*-associated episignatures, each defined by partially overlapping subsets of samples (Fig. 3). *MAU2*-specific Episignature 1 was derived from a subset of nine samples carrying six different variants (p.(Cys50Ser), p.(Asp242Glyfs*28), three individuals with p.(Gln135Argfs*32), p.(Val54_Pro56del), p.(Lys322del), and two individuals with p.(Ala378_Gln380delinsAsp)). These samples shared a consistent pattern of differential methylation relative to controls, allowing for the identification of 211 similarly differentially methylated probes (Fig. 3A–D; Supplementary Data 3). Based on this observation, a second discovery analysis was performed using a subset of five similar samples as a training set. When the remaining *MAU2* samples were tested against this model, two additional cases matched the derived profile, resulting in a second episignature consisting of seven *MAU2* samples and defined by 203 probes (Fig. 3E–H; Supplementary Data 4). Specifically, the second *MAU2*-specific episignature was defined by the following seven cases, carrying three distinct variants: three with p.(Gln135Argfs*32), two with p.(Ala378_Gln380delinsAsp), and two with p.(Ala601Leufs*124). Hence, the two *MAU2*-specific episignatures included both CdLS-positive and CdLS-negative cases. Some variants contributed to both episignatures, others to one or neither, highlighting heterogeneity in *MAU2* methylation profiles. Strikingly, in a mother–son pair with the p.(Ala601Leufs*124) variant (Individuals 11 and 12), CdLS episignature results were discordant: the mother tested positive, while the son did not, despite retaining the *MAU2*-specific episignature 1. The son's sample also showed an epigenetic age mismatch (chronological 16 years, predicted 27 years), suggesting that technical or biological factors may have affected detection of the CdLS-specific episignature in this individual.

Comparison of the probe sets underlying the two *MAU2*-specific episignatures revealed minimal overlap (14 shared probes, all hypomethylated), indicating that while the two episignatures are related, they are driven predominantly by distinct genomic regions. Each episignature captures a subset of shared methylation features, reflecting only part of the broader *MAU2*-associated methylation landscape. Variants represented in both episignatures share common differentially methylated CpGs despite differences in probe composition, whereas variants present in only one episignature exhibit methylation changes that are more consistently detected within a specific subset of *MAU2* samples. These episignature groupings may reflect biological factors such as variant-specific effects, differences in variant location, similar functional consequences, or variability in clinical phenotype.

## A subset of *MAU2* variants impairs NIPBL–MAU2 heterodimerization

MAU2 promotes proper folding and stability of NIPBL by enwrapping its N-terminal region[8]. *MAU2* variants that affect residues critical for this molecular interface may therefore compromise the integrity of the MAU2-NIPBL interaction. Supporting this hypothesis, we have shown that the previously reported MAU2 p.(Gln310_Ala316del) in-frame deletion of Individual 1 impairs this interaction[11]. Expanding on this observation, we sought to systematically assess the impact of all missense substitutions and in-frame deletions of our cohort using a quantitative mammalian two-hybrid assay, as previously described[11]. This analysis revealed that six out of ten tested *MAU2* variants significantly impaired complex formation (Fig. 4). Three in-frame deletions partially overlapped within the same region of MAU2: p.(Ala309_Lys322del), p.(Gln310_Ala316del), and p.(Lys322del). The two deletions spanning multiple amino acids markedly disrupted the interaction, whereas the p.(Lys322del) affecting the terminal end of this region had a milder, non-significant effect, indicating that both the position and extent of the deletion influence NIPBL binding efficiency. Importantly, the results of the mammalian two-hybrid assay were consistent with those of the episignature study in individuals for whom blood DNA was available (Fig. 4). Blood samples from individuals with variants that significantly disrupted the MAU2-NIPBL interaction also showed a positive CdLS episignature (p.(Val54_Pro56del), p.(Gln310_Ala316del), p.(Ala378_Gln380delinsAsp), and p.(Leu381-Pro)). Conversely, variants without significant interaction impairment (p.(Cys50Ser), p.(Lys322del), p.(Leu528Phe)) were negative for the CdLS episignature. These findings indicate that impaired MAU2–NIPBL interaction strongly aligns with the CdLS molecular signature.

Notably, two individuals with variants that did not disrupt the MAU2-NIPBL interaction (p.(Cys50Ser) and p.(Lys322del)) displayed a positive *MAU2*-specific episignature. While the luciferase assay is unable to detect which variants could potentially lead to the *MAU2*-specific episignature, this observation provides an opportunity for future studies to explore MAU2-specific molecular effects that may operate independently of NIPBL interaction.

To distinguish whether the reduced mammalian two-hybrid assay signal resulted from impaired protein–protein interaction or from variant-induced MAU2-destabilization unrelated to the interaction, we evaluated the expression of all pCMV-AD_MAU2 constructs used in the assay by Western blotting using both MAU2- and NF-κB-specific antibodies, with the corresponding endogenous protein (MAU2 or NF-κB) serving as a loading control (Supplementary Fig. 2A, B). One mutant, p.(Ala309_Lys322del), consistently showed reduced expression across replicates with both antibodies, suggesting possible protein instability in addition to impaired NIPBL interaction. To systematically evaluate the impact of *MAU2* variants on protein stability, we conducted a cycloheximide chase assay. All but one variant exhibited little to no change in stability relative to the wild-type protein, although some showed statistically significant differences with small effect size (Supplementary Fig. 2C, D). In contrast, the p.(Ala309_Lys322del) variant displayed a pronounced reduction in stability. These findings indicate that the pathogenicity of the p.(Ala309_Lys322del) variant likely arises from a combination of impaired NIPBL interaction and decreased protein stability, a factor that should be considered when interpreting the mammalian two-hybrid assay results.

## A frameshift variant leads to *MAU2* haploinsufficiency in fibroblasts

Following the analysis of in-frame variants, we next examined the functional consequences of truncating *MAU2* variants. To explore this, we analyzed primary dermal fibroblasts from carriers of the frameshift deletion p.(Gln135Argfs*32). As cells from Individual 15 failed to expand, experiments were conducted on cells of Individual 14 and her carrier father. These samples were used to assess the impact of the

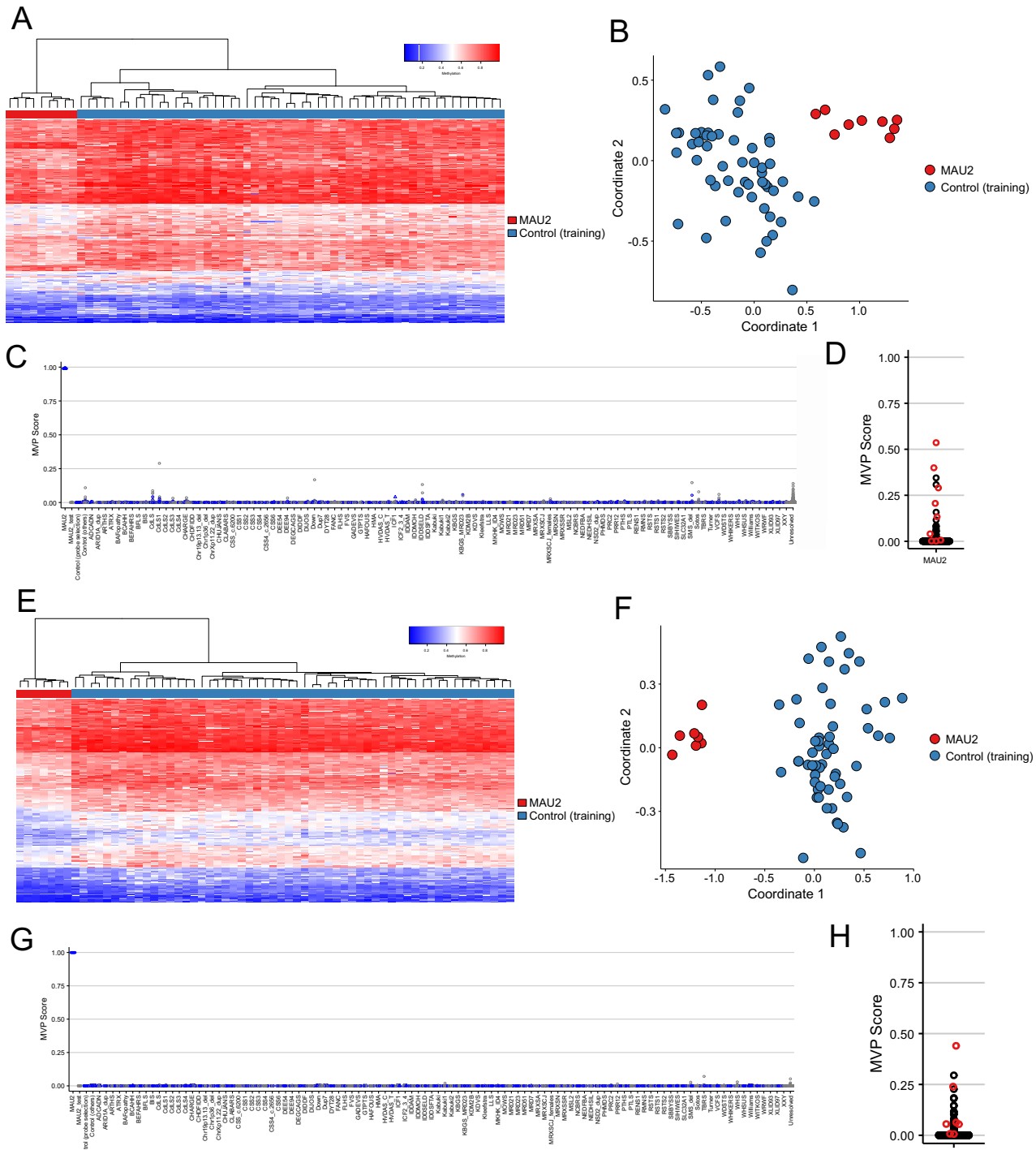

**Fig. 3 | Identification and evaluation of the *MAU2*-specific episignatures.**
**A**–**D** Characterization of the *MAU2*-specific episignature 1 using nine samples.
**E**–**H** Characterization of the *MAU2*-specific episignature 2 using seven samples.
**A, E** Euclidean hierarchical clustering (heatmaps): each column represents a single
*MAU2* sample or a matched control, and each row corresponds to one of the CpG
probes selected for episignature discovery. *MAU2* variant carriers (red) and con-
trols (blue) cluster separately, with *MAU2* samples exhibiting a methylation pattern
distinct from controls. **B, F** Multidimensional scaling (MDS) plot showing separa-
tion between *MAU2* variant carriers (red) and controls (blue). **C, G** MVP scores
calculated by the SVM classifier are plotted, with training samples shown as blue
circles and testing samples as gray circles. The model was trained using the *MAU2*

variant carriers, their matched controls, and 75% of additional controls and 75% of
other disorder samples from the EKD, while the remaining 25% of controls and 25%
of disorder samples were used as the testing set. MVP scores for *MAU2* samples are
near 1, while the remaining training and testing samples cluster near 0, indicating
specificity for the classifier. **D, H** Summary MVP plot of nine rounds (**D**) or seven
rounds (**H**) of leave-one-out cross validation (LOOCV) of the *MAU2* classifier. For
each round of LOOCV, an MVP score was generated for the withheld sample. The
MVP scores for the *MAU2* samples (red) were then plotted alongside the average
MVP scores across all LOOCV rounds for other samples in the EKD (black), including
controls and individuals with different episignature disorders.

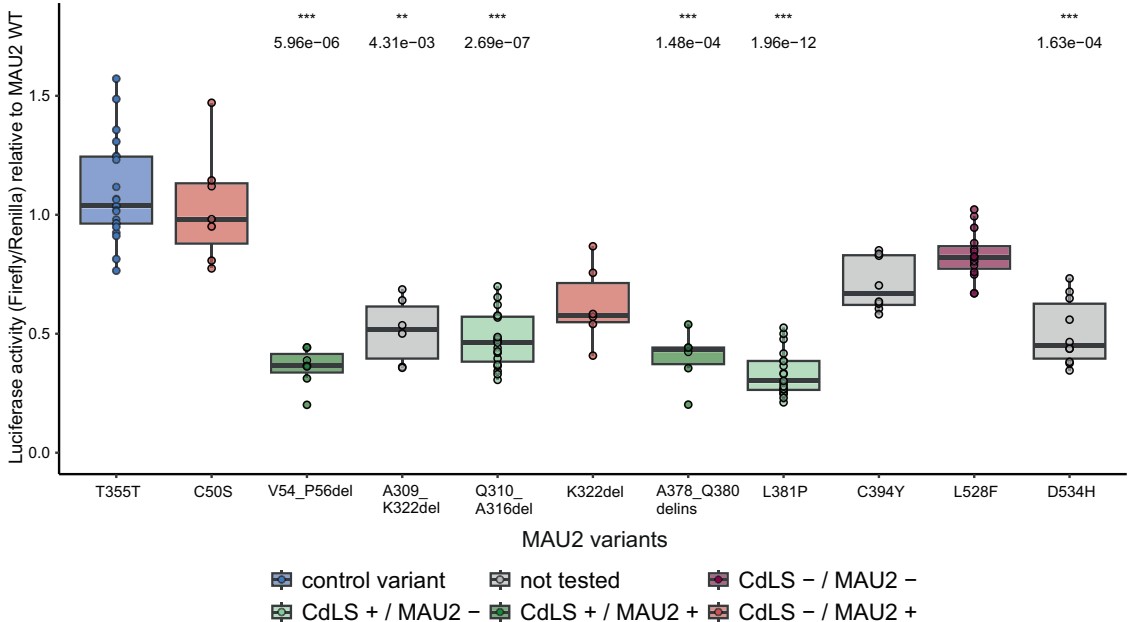

**Fig. 4 | Results of the mammalian two-hybrid assay quantifying the interaction strength between MAU2 and NIPBL.** The Firefly-to-Renilla ratio for each variant was normalized to that of the wild-type MAU2 construct, which was assigned a value of 1 in the graph. A control variant, T355T (reported as frequent in the healthy population: 625/1,612,168 alleles in gnomAD v4.1.0), was included as a negative control. The control variant T355T is highlighted in blue. All measurements were performed in three technical replicates per sample, which were averaged for statistical analysis. Each sample was analyzed in at least six independent biological experiments, with each biological replicate corresponding to a separate transfection, lysis, and luminescence measurement. Statistical significance was assessed using a two-sided Dunn's test with Bonferroni correction for multiple comparisons. Statistically significant P-values are indicated above the corresponding variant; variants lacking P-values did not achieve statistical significance (*$P \le 0.05$, **$P \le 0.01$, ***$P \le 0.001$). Box plot elements are defined as follows: center line: median; box limits: upper and lower quartiles; whiskers: 1.5× interquartile range. The mammalian two-hybrid assay results were correlated with the episignature analysis. Samples not available for episignature testing are represented in gray, those exhibiting a positive CdLS episignature and a positive MAU2 episignature in green, those exhibiting a positive CdLS episignature and a negative MAU2 episignature in light green, those exhibiting a negative CdLS episignature and a positive MAU2 episignature in red, and those with a methylation profile negative for both the CdLS and MAU2 episignature in purple. Source data are provided as a Source Data file.

frameshift variant on MAU2 transcript levels by qPCR. The analysis was extended to include Individual 1, who carries an in-frame deletion known to disrupt the MAU2–NIPBL interaction. MAU2 transcript levels were normalized to two independent housekeeping genes (NADH and SNAPIN) and compared to the mean expression levels of four unrelated healthy controls. A total of six experiments were conducted across three biological replicates. The results revealed a reduction of the MAU2 transcript levels in both Individual 14 and her father, consistently across both normalization strategies (Fig. 5A, B, Supplementary Fig. 3). In contrast, MAU2 transcript levels in fibroblasts from Individual 1 were comparable to control levels. This indicates that the frameshift deletion p.(Gln135Argfs*32), but not the in-frame variant p.(Gln310_Ala316del), leads to reduced MAU2 transcript abundance, supporting nonsense-mediated mRNA decay (NMD)–induced haploinsufficiency as the likely pathogenic mechanism. We next analyzed NIPBL transcript levels, which showed no consistent changes across samples (Fig. 5C, D, Supplementary Fig. 3). However, a reduction of NIPBL at the protein level may still occur due to the mutual dependence between NIPBL and MAU2 for stability. Western Blot analysis of total lysates supports reduced abundance of both MAU2 and NIPBL proteins in Individual 14 and her father (Fig. 5E, Supplementary Fig. 4), suggesting that MAU2 haploinsufficiency may extend to NIPBL reduction at the protein level.

### MAU2 variants result in variable phenotypes marked by short stature and microcephaly

Our cohort of individuals with MAU2 variants comprises nine males and nine females, ranging in age from 20 months to 39 years (median age 12 years). Comprehensive phenotypic data for each individual are provided in Supplementary Data 5. The core clinical features observed in our cohort are summarized in Table 1. The most frequently reported phenotypes include short stature (83%), microcephaly (75%), intellectual disability (61%), synophrys (56%), long philtrum (56%), thick eyebrows (50%), smooth philtrum (50%), thin upper lip vermilion (44%), central nervous system anomalies (42%), anteverted nostrils (39%), behavioral abnormalities (38%), hypertrichosis (38%), speech delay (35%), genitourinary anomalies (27%), and downturned corners of the mouth (22%) (Fig. 6A). These observations indicate that short stature and microcephaly are the two major clinical features associated with MAU2 variants. Both were often already present at birth, with each observed in approximately 50% of individuals for whom neonatal data were available (microcephaly in 5/11 and short stature in 8/16 individuals). Six of the 15 individuals with short stature received growth hormone (GH) treatment (Individuals 2, 5, 10, 14, 15, and 17) (Supplementary Data 5). The treatment began as early as 1 month (Individual 5) and as late as 13 years (Individual 2). Although several individuals were treated for up to ten years, improvement in height trajectory was sporadic and not observed consistently across all individuals.

Intellectual disability (ID) was the third most common feature, although it varied in severity among individuals. The majority exhibited learning difficulties (4/11) or borderline-to-mild ID (4/11), while three individuals showed moderate-to-severe ID (Fig. 6B). Thus, cognitive impairment in these individuals appears to be generally mild. Behavioral anomalies were reported in fewer than half of the cases, with attention-deficit/hyperactivity disorder (ADHD), hyperactivity, and anxiety being the most common features. Developmental milestones were achieved in the majority of cases within the expected age range, with a median age of independent walking at 15 months and first

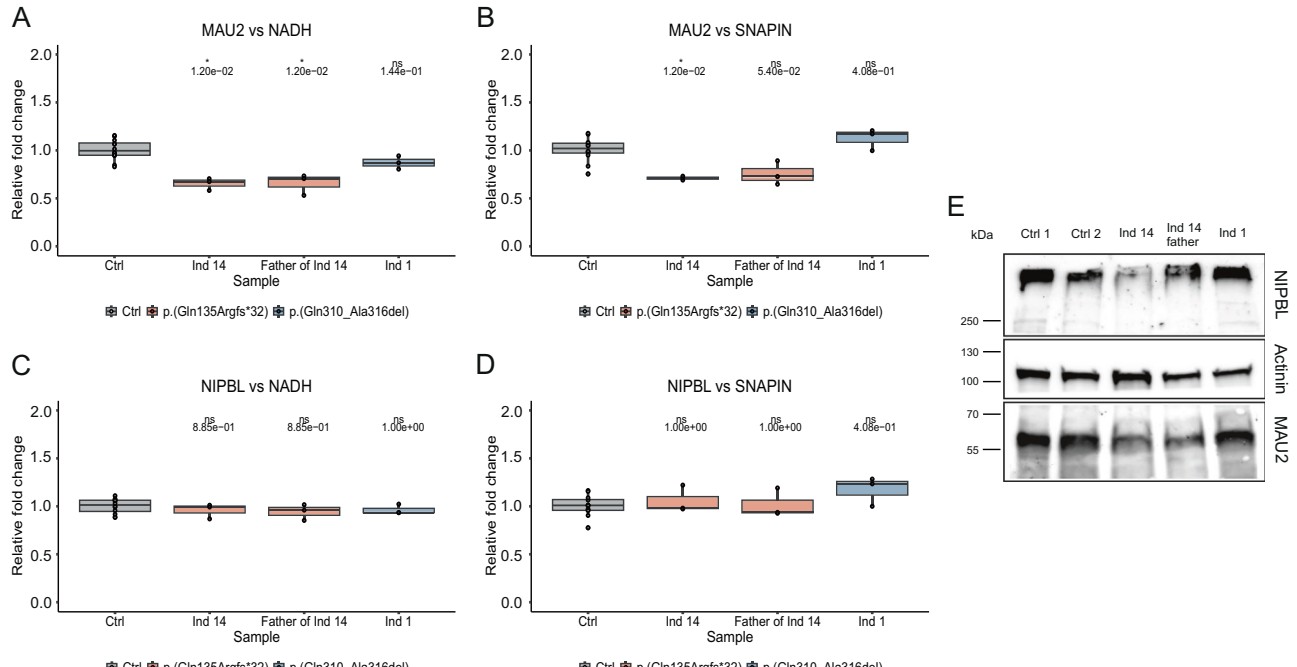

**Fig. 5 | Fibroblasts with a frameshift variant exhibit *MAU2* haploinsufficiency.**
**A–D** Quantification of *MAU2* and *NIPBL* transcript levels by qPCR. In all panels, data from four unrelated healthy controls are represented in gray, samples harboring the frameshift variant p.(Gln135Argfs*32) are shown in red, and Individual 1 with the in-frame variant known to disrupt NIPBL interaction (p.(Gln310_Ala316del)) is shown in blue. Six independent experiments were performed, with two technical replicates for each of the three RNA preparations. Technical replicates were averaged to yield one value per biological replicate, which was used for statistical comparisons (two-sided Mann–Whitney U test followed by Bonferroni Correction for multiple comparisons, * $P \leq 0.05$, ** $P \leq 0.01$, *** $P \leq 0.001$). *P*-values are indicated above the corresponding box plot. Box plot elements are defined as follows: center line: median; box limits: upper and lower quartiles; whiskers: $1.5\times$ interquartile range. (A) *MAU2* transcript levels normalized to *NADH*. **B** *MAU2* transcript levels normalized to *SNAPIN*. **C** *NIPBL* transcript levels normalized to *NADH*. **D** *NIPBL* transcript levels normalized to *SNAPIN*. **E** Assessment of MAU2 and NIPBL protein levels by Western Blot. Protein levels were assessed in two healthy controls (Ctrl 1 and Ctrl 2), Individual 14 and her father (*MAU2* frameshift variant), and in Individual 1 (*MAU2* in-frame deletion). The membranes were cut to allow all three proteins (NIPBL, MAU2, and the loading control Actinin) to be visualized from a single blot. This blot is representative of three independent experiments with similar results. Source data are provided as a Source Data file.

spoken words at 14 months. Brain MRI abnormalities were also frequently observed, although the findings were largely nonspecific (Supplementary Data 5).

Molecular evidence supporting a diagnosis of CdLS was obtained in 11/18 cases (Table 1), either via a positive CdLS episignature or significant disruption of the NIPBL–MAU2 interaction. Three additional individuals showed only a *MAU2*-specific episignature, suggesting a milder or variant-specific effect, while two tested negative across all molecular assays conducted in this study. Appropriate testing was not possible for the remaining two individuals due to sample unavailability (Table 1). To explore whether individuals with molecularly confirmed CdLS exhibit a recognizable clinical profile, we compared the frequency of clinical features observed across the entire cohort with that in the subset of confirmed CdLS cases (defined by a positive CdLS episignature or impaired NIPBL–MAU2 interaction). While microcephaly and short stature were comparably frequent across both groups, facial dysmorphism was more common among individuals with molecularly confirmed CdLS (Fig. 6A, C). The features with a higher frequency in the CdLS-subgroup included thick eyebrows, synophrys, anteverted nostrils, long and smooth philtrum, thin upper lip vermilion, and downturned corners of the mouth. These characteristic facial features can be recognized in the facial photographs of the individuals who consented to publication (Fig. 7). The distribution of ID severity was instead comparable to that observed in the full cohort (Fig. 6B, D).

To delineate the clinical specificity of *MAU2*-associated CdLS, we next compared the frequency of canonical CdLS features (as defined by Kline et al.[1]) across five groups: (i) individuals with pathogenic *NIPBL*

variants, (ii) *MAU2* variant carriers with molecularly confirmed CdLS, (iii) non-CdLS *MAU2* variant carriers, (iv) *MAU2* variant carriers for whom functional or episignature analyses were not possible, (v) *MAU2* variant carriers that yielded negative results in all molecular and functional analyses conducted here (Table 2). Although facial dysmorphism is more frequent in the molecularly confirmed *MAU2*-CdLS subgroup than in the overall *MAU2* cohort (Fig. 6A, C), the gestalt remains generally subtler than in *NIPBL*-related CdLS. While arched/thick eyebrows and synophrys occur at comparable frequencies, *MAU2*-CdLS cases show less involvement of perioral and nasal structures, including a lower prevalence of thin upper vermilion, downturned corners of the mouth, and depressed nasal bridge. Limb anomalies are also milder, with no major malformations and fewer instances of small hands/feet or proximally placed thumbs. Intellectual disability and developmental delay likewise tend to be less frequent and considerably milder than in *NIPBL*-CdLS. Hearing impairment follows a similar trend, being observed in two of 18 individuals in our cohort compared with the estimated prevalence of 85–90% in classical CdLS[1]. Overall, these comparisons indicate that *MAU2*-associated CdLS presents with an attenuated phenotype, with no individual feature or combination of features that would allow a clinician to predict the presence of a *MAU2* variant based on clinical examination alone.

Given the association between *MAU2* and CdLS emerging from our data, we applied the published CdLS clinical scoring system to all individuals in the cohort (Table 1, Supplementary Data 6). Three individuals met criteria for classic CdLS, each scoring 13 points (Individuals 1, 2, and 14). Six individuals fell within the non-classic CdLS range (9–10 points) (Individuals 4, 10, 11, 15, 16, and 17), while five scored between 4

**Table 1 | Summary of the clinical features and experimental results**

| Individual number | Ind 1 | Ind 2 | Ind 3 | Ind 4 | Ind 5 | Ind 6 | Ind 7 | Ind 8 | Ind 9 | Ind 10 | Ind 11 | Ind 12 | Ind 13 | Ind 14 | Ind 15 | Ind 16 | Ind 17 | Ind 18 |
|---|---|---|---|---|---|---|---|---|---|---|---|---|---|---|---|---|---|---|
| MAU2 variant | p.(Gln310_Ala316del) | p.(Leu381Pro) | p.(Leu528Phe) | p.(Asp534His) | p.(Cys394Tyr) | p.(Cys505Ser) | p.(Asp242Glyfs*28) | p.(Cys145Leufs*14) | p.(Val54_Pro56del) | p.(Gln538*) | p.(Ala601Leufs*124) | p.(Ala601Leufs*124) | p.(Lys322del) | p.(Gln135Argfs*32) | p.(Gln135Argfs*32) | p.(Ala378_Gln380delinsAsp) | p.(Ala378_Gln380delinsAsp) | p.(Ala309_Lys322del) |
| Inheritance | de novo | und | de novo | de novo | de novo | paternal | und | und | de novo | de novo | maternal | und | de novo | paternal | paternal | maternal | de novo | und |
| CdLS episignature | yes | yes | no | und | und | no | yes | und | yes | und | no | yes | no | yes | yes | yes | yes | und |
| MAU2 episignature | no | no | no | und | und | yes | yes | und | yes | und | yes | yes | yes | yes | yes | yes | yes | und |
| Experimental consequences | impaired NIPBL interaction | impaired NIPBL interaction | none detected | impaired NIPBL interaction | none detected | none detected | und | und | impaired NIPBL interaction | und | und | und | none detected | MAU2 haploinsufficiency | und | impaired NIPBL interaction | impaired NIPBL interaction | impaired NIPBL interaction |
| CdLS clinical score | 13 | 13 | 4 | 10 | 4 | 3 | 6 | 2 | 7 | 10 | 9 | 5 | 4 | 13 | 10 | 10 | 10 | 8 |
| Short stature | yes | yes | no | no | yes | yes | yes | yes | no | yes | yes | yes | yes | yes | yes | yes | yes | yes |
| Microcephaly | yes | yes | yes | yes | no | NA | NA | no | yes | yes | yes | yes | yes | yes | yes | no | yes | no |
| Thick eyebrows | yes | yes | no | no | no | no | yes | no | no | yes | yes | no | no | yes | yes | yes | yes | no |
| Synophrys | yes | yes | no | yes | no | no | no | no | no | no | no | no | yes | no | no | no | no | yes |
| Depressed nasal bridge | yes | yes | no | NA | no | no | no | no | no | no | no | no | no | no | no | no | no | no |
| Anteverted nostrils | yes | yes | no | no | no | no | yes | no | yes | no | no | no | no | yes | yes | yes | no | no |
| Long philtrum | yes | yes | no | yes | no | yes | no | no | yes | no | no | no | no | yes | yes | yes | yes | yes |
| Smooth philtrum | yes | yes | no | no | no | yes | no | no | no | yes | no | no | no | yes | yes | yes | yes | yes |
| Thin upper lip vermilion | yes | yes | no | yes | no | no | no | no | no | no | yes | no | no | yes | no | yes | yes | yes |
| Downturned corners of the mouth | yes | yes | no | yes | no | no | no | no | no | no | no | no | no | no | no | no | no | yes |
| Structural eye anomalies | no | no | yes | no | yes | NA | NA | no | no | no | no | no | no | no | no | no | no | no |
| Hypertrichosis | yes | yes | no | yes | no | NA | NA | no | no | yes | no | no | no | yes | yes | no | no | no |
| Cardiac anomalies | no | no | no | no | NA | NA | NA | no | NA | yes | NA | NA | no | no | no | yes | NA | no |
| Genitourinary anomalies | yes | no | no | no | yes | no | no | no | NA | no | NA | NA | no | yes | yes | no | NA | no |
| CNS anomalies | yes | no | no | yes | yes | NA | NA | no | no | no | NA | NA | no | yes | yes | NA | no | no |
| Feeding problems | yes | no | no | no | yes | no | no | no | no | no | no | NA | no | yes | no | no | no | no |
| ID | yes | yes | yes | no | yes | yes | yes | yes | yes | yes | yes | yes | no | no | no | no | no | yes |
| Behavioral issues | no | no | no | no | yes | NA | yes | yes | yes | no | yes | yes | no | no | no | no | no | NA |
| Motor delay | yes | no | yes | no | yes | no | no | NA | no | no | no | NA | no | no | no | no | no | no |
| Speech delay | yes | no | yes | yes | yes | no | no | yes | yes | no | no | NA | no | no | no | no | no | no |

*und* undetermined, *ID* intellectual disability, *CNS* central nervous system, *NA* not available.

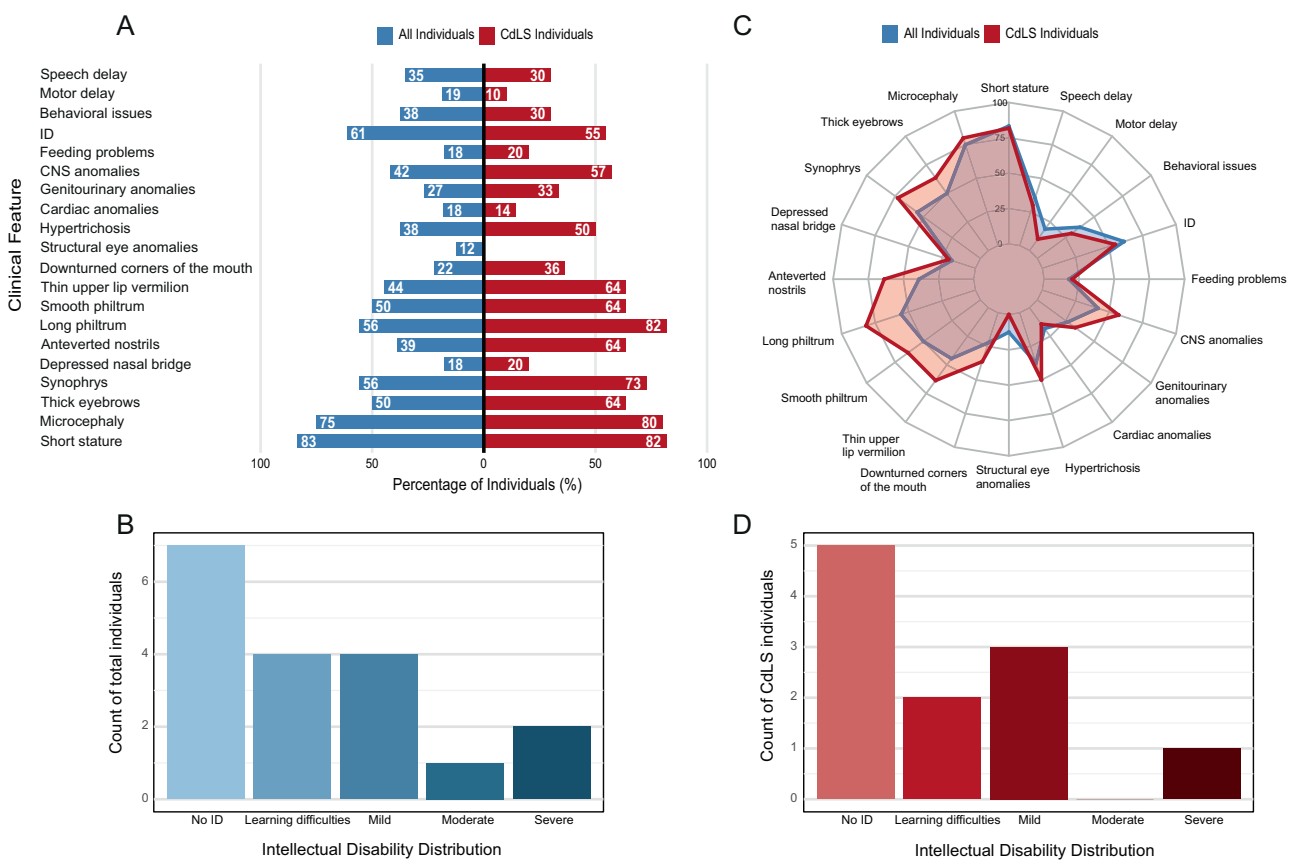

**Fig. 6 | Frequency of key clinical features associated with *MAU2* variants.**
**A** Mirror plot illustrating the frequency (as a percentage) of the main clinical features in the total cohort (blue) compared to molecularly confirmed CdLS individuals (red). **B** Distribution of intellectual disability (ID) severity within the total cohort. **C** Radar plot emphasizing the increased prevalence of facial dysmorphism in molecularly confirmed CdLS individuals. **D** Distribution of ID severity within the molecularly confirmed CdLS cohort.

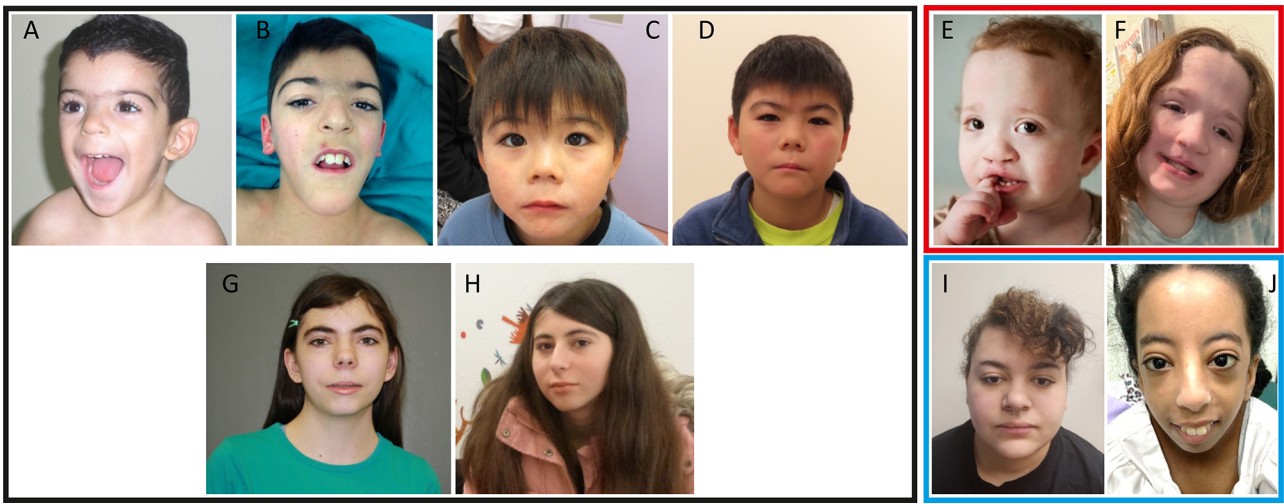

**Fig. 7 | Facial appearance of individuals with *MAU2* variants.** Facial photographs of individuals with molecularly confirmed CdLS are outlined in black. Those whose variants did not show any effect in the mammalian two-hybrid assay are outlined in red, while those without available material for molecular analysis are outlined in blue. **A**, **B** Individual 1, at 2 years and 10 years of age. CdLS clinical score: 13. **C**, **D** Individual 2 at 5 years and 11 years of age. CdLS clinical score: 13. **E**, **F** Individual 5, at 23 months and 12 years of age. CdLS clinical score: 4. **G** Individual 14 at 22 years of age. CdLS clinical score: 13. **H** Individual 15 at 16 years of age. CdLS clinical score: 10. **I** Individual 8 at 19 years of age. CdLS clinical score: 2. **J** Individual 10 at 14 years of age. CdLS clinical score: 10.

**Table 2 | Phenotype comparisons across different subtypes**

| | HPO ID | *NIPBL*-CdLS | *MAU2*-CdLS | *MAU2*-not CdLS | *MAU2*-undetermined | *MAU2*-negative |
|---|---|---|---|---|---|---|
| | | Kline et al. | Ind. 1,2,4,7,9,12,14,15,16,17,18 | Ind. 6, 11, 13 | Ind. 8, 10 | Ind. 3, 5 |
| **Growth** | | | | | | |
| IUGR | HP:0001511 | +++ | + (5/11) | + (1/3) | ++++ (1/1) | ++++ (1/1) |
| Short stature | HP:0004322 | +++ | +++ (9/11) | ++++ (3/3) | ++++ (2/2) | ++ (1/2) |
| Microcephaly | HP:0000252 | ++++ | +++ (8/10) | ++++ (2/2) | ++ (1/2) | ++ (1/2) |
| **Craniofacial features** | | | | | | |
| Brachycephaly | HP:0000248 | ++ | – (2/11) | – (0/3) | ++ (1/2) | – (0/2) |
| Low anterior hairline | HP:0000294 | +++ | + (5/11) | – (0/3) | ++ (1/2) | – (0/2) |
| Arched, thick eyebrows | HP:0002253, HP:0000574 | +++ | +++ (9/11) | + (1/3) | ++ (1/2) | – (0/2) |
| Synophrys | HP:0000664 | ++++ | +++ (8/11) | ++ (2/3) | – (0/2) | – (0/2) |
| Long eyelashes | HP:0000527 | ++++ | ++ (6/10) | – (0/2) | ++ (1/2) | ++ (1/2) |
| Depressed nasal bridge | HP:0005280 | +++ | + (2/10) | – (0/3) | – (0/2) | ++ (1/2) |
| Anteverted nostrils | HP:0000463 | +++ | ++ (7/11) | – (0/3) | – (0/2) | – (0/2) |
| Broad nasal tip | HP:0000455 | ++ | – (1/11) | – (0/3) | ++++ (2/2) | ++++ (2/2) |
| Long, smooth philtrum | HP:0000343, HP:0000319 | +++ | +++ (9/11) | + (1/3) | ++ (1/2) | – (0/2) |
| Thin upper vermilion | HP:0000219 | ++++ | ++ (7/11) | + (1/3) | – (0/2) | – (0/2) |
| Downturned corners of the mouth | HP:0002714 | ++++ | + (4/11) | – (0/3) | – (0/2) | – (0/2) |
| Widely spaced teeth | HP:0000687 | +++ | – (1/10) | – (0/3) | – (0/2) | – (0/2) |
| Low-set and mal-formed ears | HP:0000369, HP:0000377 | ++ | – (1/11) | – (0/3) | – (0/2) | ++++ (2/2) |
| **Trunk and limbs** | | | | | | |
| Oligodactyly and adactyly (hands) | HP:0012165, HP:0009776 | + | – (0/11) | – (0/2) | – (0/2) | – (0/2) |
| Small hands | HP:0200055 | +++ | + (5/11) | ++ (1/2) | ++ (1/2) | ++ (1/2) |
| Proximally placed thumbs | HP:0009623 | ++ | – (2/11) | – (0/2) | – (0/2) | ++ (1/2) |
| Clinodactyly or short fifth finger | HP:0004209, HP:0009237 | +++ | ++ (5/10) | – (0/2) | – (0/2) | ++ (1/2) |
| Small feet | HP:0001773 | ++++ | + (5/11) | – (0/2) | ++ (1/2) | ++ (1/2) |
| Hirsutism | HP:0001007 | +++ | ++ (5/10) | – (0/2) | ++ (1/2) | – (0/2) |
| **Cognition and behavior** | | | | | | |
| Intellectual disability (any degree) | HP:0001249 | ++++ | ++ (6/11) | + (1/3) | ++++ (2/2) | ++++ (2/2) |
| Motor delay | HP:0001270 | delayed | – (1/10) | – (0/3) | – (0/1) | ++++ (2/2) |
| Speech delay | HP:0000750 | delayed | + (3/10) | – (0/3) | ++ (1/2) | ++++ (2/2) |
| ASD | HP:0000729 | + | – (0/10) | – (0/2) | – (0/2) | – (0/2) |
| Self-injurious behavior | HP:0100716 | +++ | – (0/10) | – (0/2) | – (0/2) | ++ (1/2) |
| Stereotypic movements | HP:0000733 | ++ | – (0/10) | – (0/2) | – (0/2) | – (0/2) |

*Ind* Individual, *ASD* autism spectrum disorder, *HPO ID* Human Phenotype Ontology Identifier, *IUGR* Intrauterine growth retardation. ++++: >90%. +++: 70-89%. ++: 50-69%. +: 20-49%. -: <20%.

and 8 points and presented with at least one cardinal feature, thus meeting the criteria for molecular testing for CdLS (Individuals 3, 7, 9, 13, and 18). The remaining four individuals scored below 4, or above 4 without any cardinal features, thus falling outside the diagnostic threshold for CdLS (Individuals 5, 6, 8, and 12). Overall, individuals with scores ≥9 (classic and non-classic CdLS) displayed facial or clinical features overlapping with the CdLS spectrum, whereas none of the individuals scoring <9 exhibited facial features indicative of CdLS (Fig. 7). Remarkably, all three individuals classified as classic CdLS belonged to the molecularly confirmed CdLS group. Among the six individuals with non-classic CdLS scores, four had molecular confirmation (Individuals 4, 15, 16, and 17). Of the remaining two, one exhibited a *MAU2*-specific episignature (Individual 11), while the other could not be tested due to unavailability of samples (Individual 10). On the other hand, none of the two individuals with negative results in

both episignature profiling and protein interaction assays (Individuals 3 and 5) exceeded the diagnostic threshold for CdLS (≥5 points). This concordance between clinical scoring and molecular data supports the reliability and specificity of the molecular assays used in this study.

Importantly, intrafamilial differences were observed in some cases. For instance, whereas Individuals 16 and 17 (index and his mother) both scored 10, Individual 11 scored 9, while his mother (Individual 12) scored 5; Individual 14 scored 13, whereas her sister (Individual 15) scored 10.

**Heterozygous *Mau2* knockout mice show decreased body length, microcephaly, and brain anomalies**
To investigate whether the consequences of *Mau2* loss-of-function in vivo mirror the phenotypic features observed in our cohort, we generated a constitutive *Mau2* knockout mouse line in collaboration

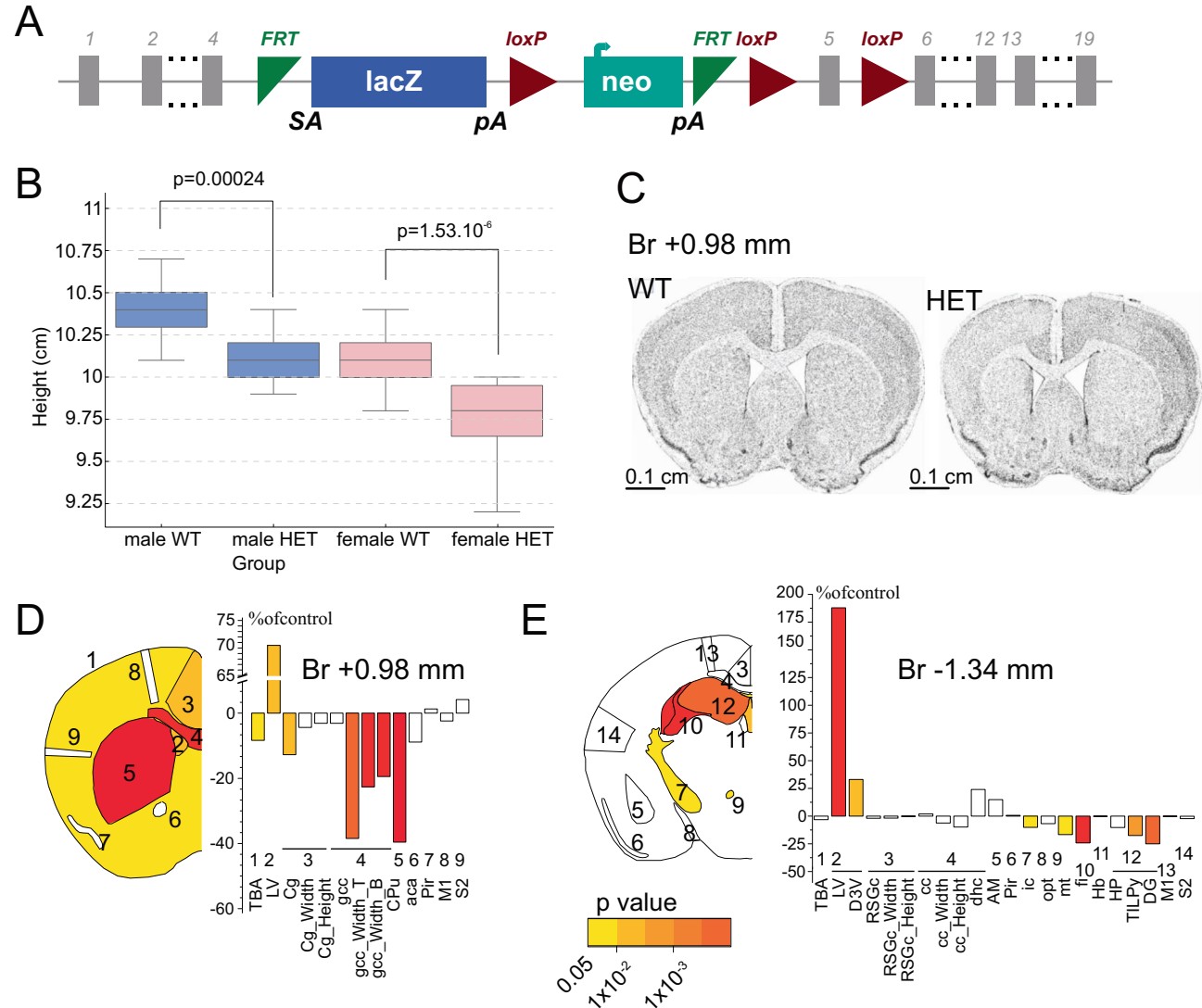

**Fig. 8 | Neuroanatomical defects in the *Mau2*⁺/⁻ mouse. A** Tm1a genetic construct showing the insertion of the LacZ and Neo cassettes upstream the critical exon 5 of *Mau2*. **B** Body length boxplots showing the median (center line), interquartile range (box), and range excluding outliers (whiskers). Statistical comparison was performed with a two-tailed Student's *t* test assuming equal variances. Individual data points are overlaid. Males are shown in blue, females in pink. Red markers represent the group mean ± standard error of the mean (SEM). Groups are displayed as: Male wild-type (WT) (*n* = 345), Male heterozygous (HET) (*n* = 7), Female WT (*n* = 340), Female HET (*n* = 7). **C** Representative coronal brain sections from *Mau2*⁺/⁻ mouse (left) and matched WT (right), showing the microcephaly phenotype at Bregma (Br) + 0.98 mm. Scale bar: 0.1 cm. Statistical comparison was performed with a two-tailed Student's *t* test assuming equal variances. **D** Schematic representation of affected brain regions at Bregma +0.98 mm according to *P*-values in three adult male *Mau2*⁺/⁻ mice aged 16 weeks compared to matched WT mice (*n* = 44). White indicates not significant and gray not computable. Histograms represent percentage change relative to WT mice (set as 0) for each of the measured parameters. Numbers indicate a group of studied brain regions. Brain structures assessed: total brain area (TBA); lateral ventricles (LV); cingulate cortex (Cg); genu of the corpus callosum (gcc); caudate putamen (CPu); anterior commissure (aca); piriform cortex (Pir); primary motor cortex (M1); secondary somatosensory cortex (S2). Data were analyzed by two-tailed Student's *t*-tests assuming equal variances. **E** Schematic representation of affected brain regions at Bregma −1.34 mm, analysed as in (**D**). Histograms represent percentage change relative to WT. Brain structures assessed: the total brain area (TBA); lateral- (LV) and third ventricles (D3V); retrosplenial granular cortex (RSGc); corpus callosum (cc); dorsal hippocampal commissure (dhc); amygdala (AM); piriform cortex (Pir); internal capsule (ic); optic tract (opt); mammillothalamic tract (mt); fimbria (fi); habenular nucleus (Hb); hippocampus (HP); total internal length of pyramidal cells (TILpy); dentate gyrus (DG); primary motor cortex (M1); secondary somatosensory cortex (S2). Data were analyzed by two-tailed Student's *t*-tests assuming equal variances.

with the Sanger Institute Mouse Genetics Project[17–19]. The *Mau2*^tm1b(KOMP)Wtsi mice were generated using the knockout-first allele method[20] developed by the International Mouse Phenotyping Consortium (IMPC), in which exon 5 of *Mau2* is floxed and a LacZ cassette inserted (Fig. 8A).

Mouse survival was assessed from 102 successfully genotyped mice derived from a heterozygous-by-heterozygous ("het-by-het") breeding scheme. The observed genotypic distribution was 0% homozygous, 57% heterozygous, and 43% wild-type (Supplementary Data 7), indicating that the complete loss of *Mau2* is incompatible with

life, consistent with a previous report[21]. Since only heterozygous mice could be generated, the *Mau2*^tm1b(KOMP)Wtsi mouse model will be hereafter referred to as *Mau2*⁺/⁻. We recovered more heterozygous females than males (37 vs. 21), and this sex imbalance suggests a previously unreported sub-viability associated with the heterozygous state that is specific to males (Supplementary Data 7). A one-tailed binomial test confirmed that the excess of females was significant (*P* = 0.024).

Surviving mice were assessed for their body weight and body length (from the nose to the beginning of the tail) at 16 weeks of age (Supplementary Data 8). Although no significant difference was found

in body weight, the length of the mice was consistently reduced by 4% in female heterozygous ($P = 1.53E\text{-}06$) and by 3% in male heterozygous ($P = 0.00024$) mice (Fig. 8B).

We next assessed brain anatomy in 16-week-old $Mau2^{+/-}$ mice using a standardized high-resolution pipeline that measures 36 unique parameters across 23 brain regions (Supplementary Data 9), following our previously established protocols[22–24]. At Bregma +0.98 mm, the brain size in $Mau2^{+/-}$ mice was reduced by 10% ($P = 0.0168$), indicative of moderate microcephaly (Fig. 8C). We also noted a clear left-right asymmetry of the brain, particularly evident in the caudate putamen and lateral ventricles at Bregma +0.98 mm (Fig. 8C). Compared to wild-type mice, $Mau2^{+/-}$ mice exhibited a total of ten significantly altered brain regions. At Bregma +0.98 mm, these included the lateral ventricles, the cingulate cortex, the caudate putamen, the genu of the corpus callosum, and the total brain area (Fig. 8D). At Bregma −1.34 mm, significant changes were observed in the internal capsule, the hippocampal formation, the dorsal third ventricle, the mammillothalamic tract, and the fimbria of the hippocampus (Fig. 8E). While the ventricles showed enlargement, all other affected brain regions were reduced in size. For example, the cingulate cortex was reduced by 13% ($P = 0.0065$), the internal capsule by 10.1% ($P = 0.026$), the total length of pyramidal cell layer by 17.5% ($P = 0.0018$), and the height of the genu of the corpus callosum by 19.5% ($P = 4.72E\text{-}05$) (Fig. 8D, E, Supplementary Data 10). These analyses indicate that the microcephaly phenotype does not reflect equally small brain regions, but instead represents a mosaic pattern of region-specific changes. Taken together, $Mau2^{+/-}$ heterozygous knockout mice exhibit short stature, microcephaly, and region-specific neuroanatomical anomalies, recapitulating key clinical features of $MAU2$-associated phenotypes observed in this cohort.

## Discussion

Prior to this study, the disease-causing potential of $MAU2$ variants was unclear. Only a few cases had been reported, including two de novo in-frame variants and three de novo microdeletions at 19p13.11–p12 associated with broad, nonspecific neurodevelopmental phenotypes[11–13]. The large size of these deletions (1.4–5.2 Mb) and the involvement of numerous additional genes (37–68 genes) precluded definitive attribution of the observed phenotypes to $MAU2$ alone. Moreover, the scarcity of reported variants had limited our understanding of the phenotypic spectrum and underlying pathogenic mechanisms connected to $MAU2$ dysfunction. Our characterization of 18 individuals with 15 distinct $MAU2$ variants directly addresses these knowledge gaps, providing robust evidence for its pathogenic role and advancing our understanding of the associated clinical features.

To explore the molecular consequences of $MAU2$ loss-of-function variants, we analyzed fibroblasts carrying a frameshift variant. These cells displayed significantly reduced $MAU2$ mRNA levels, likely resulting from degradation of the mutant transcript via nonsense-mediated decay (NMD). This supports a haploinsufficiency mechanism at the RNA level for $MAU2$ loss-of-function variants, similar to what has been previously observed for $NIPBL$ variants[25]. In contrast, $NIPBL$ transcript levels remained unaffected in the $MAU2$ mutant cells, indicating the absence of a direct transcriptional feedback regulation between the two cohesin loader subunits. However, protein analysis suggested reduced levels of both MAU2 and NIPBL, consistent with prior siRNA-based studies showing that loss of one cohesin loader subunit destabilizes the other[9]. Similar effects were observed in induced pluripotent stem cells with $NIPBL$ variants[26]. Although our analysis was limited to a single family, it is plausible that other loss-of-function $MAU2$ variants might similarly result in $MAU2$ haploinsufficiency.

Given the interdependence between MAU2 and NIPBL, in-frame $MAU2$ variants that weaken this interaction would be expected to similarly reduce protein stability of both MAU2 and NIPBL. However, this anticipated effect was not observed in our experiments: despite substantial impairment of the interaction, protein levels remained stable in fibroblasts carrying an in-frame deletion. This may reflect a combination of dosage sensitivity and compensatory mechanisms. Functional assays showed that in-frame variants preserve partial MAU2–NIPBL binding in vitro (at least 30%), allowing some complex formation. In addition, as previously reported for $NIPBL$ variants, upregulation of the wild-type allele may further compensate, preventing major reductions in protein levels[25]. Together, these factors likely result in a greater amount of functional MAU2 protein in cells with in-frame variants compared to variants leading to $MAU2$ haploinsufficiency. Moreover, even a weakened interaction may be sufficient to maintain the stability of NIPBL and MAU2 and prevent their degradation, although it may not fully sustain downstream cohesin loader functions, such as interaction with additional transcription factors.

Reflecting a shared pathogenic mechanism characterized by partial impairment of cohesin loader function, both in-frame and truncating $MAU2$ variants lead to a DNA methylation profile that overlaps with the established CdLS episignature[16,27–29], providing strong molecular evidence linking $MAU2$ variants to CdLS. Importantly, we could also establish a novel $MAU2$-specific episignature. Interestingly, three individuals do not display the CdLS episignature but are positive for the $MAU2$-specific episignature, indicating that certain variants may partially impair cohesin loader function in a way that is sufficient to generate a $MAU2$-specific methylation signature, yet insufficient to trigger the full CdLS DNA methylation profile. Consistently, in-frame variants linked only to the $MAU2$-specific episignature did not significantly disrupt the NIPBL-MAU2 interaction in the mammalian two-hybrid assay. Clinically, these attenuated molecular effects are reflected in mild phenotypes: two of these three cases (Individuals 6 and 13) exhibited mainly short stature and subtle dysmorphic features. Variants with similarly modest functional consequences may therefore go clinically unrecognized, potentially explaining the presence of the p.(Cys50Ser) variant in gnomAD.

In our cohort, the most common phenotypic features associated with $MAU2$ variants were short stature and microcephaly, often present from birth. Six individuals with short stature received growth hormone (GH) treatment. GH deficiency was confirmed in three cases, absent in one, and unknown in two (Supplementary Data 5). Treatment was initiated between 1 month and 13 years of age, with durations up to 10 years, but generally yielded limited benefit. A direct comparison between two siblings highlighted a possible potential benefit: Individual 15, who had GH deficiency and started treatment earlier than her sister (Individual 14, who does not present with GH deficiency), exhibited milder short stature (− 2.02 SD vs. −2.97 SD) and showed height improvement from −2.29 to −2.02 SD during the last year of treatment. GH therapy remains a largely unexplored area, even among individuals with classic CdLS, where GH secretion is typically normal. Nevertheless, modest benefit has been reported even in the absence of GH deficiency[30]. In one previously reported case, a CdLS individual without GH deficiency or insensitivity initiated GH treatment at 4.3 years of age and achieved a substantial height gain of 1.6 SD over eight years, with no significant adverse reaction[31]. Given the limited number of treated individuals, GH therapy may be considered on a case-by-case basis in $MAU2$ variant carriers with significant short stature, particularly if GH deficiency is documented, but its efficacy remains uncertain.

Most individuals with $MAU2$ variants achieved developmental milestones within the expected age range, in contrast to the more pronounced delays typically observed in individuals with $NIPBL$ variants[1]. Furthermore, ID was generally milder in the $MAU2$ group compared to non-$MAU2$ CdLS cases[1], and some individuals in our cohort did not display any type of cognitive impairment. Systematic comparisons between $NIPBL$-CdLS and $MAU2$-CdLS indicate that the milder clinical presentation in $MAU2$ variant carriers extends beyond cognitive outcomes. We speculate that this discrepancy reflects

primarily quantitative and context-dependent effects. First, the reduction of NIPBL secondary to *MAU2* haploinsufficiency may be less pronounced or less likely to cross critical developmental thresholds than the primary *NIPBL* haploinsufficiency resulting from a direct *NIPBL* pathogenic variant. Consistent with this, lower *NIPBL* expression levels have been shown to correlate with worse phenotypic outcomes[25]. Second, MAU2 and NIPBL appear to have partially distinct functional contributions within the cohesin loader. NIPBL appears to provide the principal DNA-binding and cohesin-engaging interfaces[32], whereas MAU2 has been proven indispensable for proper transcriptional regulation[11]. Accordingly, fibroblasts with *MAU2* or *NIPBL* variants show a strong but incomplete correlation in dysregulated gene expression ($R = 0.79$)[11], indicating disruption of a largely shared yet not identical regulatory program. Similar gene-specific expressivity is well documented within the cohesin ring itself, where pathogenic variants in *SMC1A*, *SMC3* or, *RAD21* result in distinct phenotypes, generally attributed to subunit-specific biochemical roles and allele-dependent quantitative effects. Taken together, these quantitative differences in *NIPBL* reduction, combined with subunit-specific contributions and tissue-specific sensitivity to cohesin loader activity, likely determine whether molecular perturbations manifest as classical, multisystem CdLS, or as a milder phenotype.

Consistent with this model, clinical expressivity also appears to be variable among individuals with *MAU2* variants, as illustrated by the few familial cases in our cohort. However, penetrance cannot be reliably estimated, as clinical information is incomplete for two carrier parents and, in cases with extremely mild manifestations, it is difficult to distinguish incomplete from complete penetrance with markedly attenuated expressivity. A particularly informative familial case involves two sisters (Individuals 14 and 15) carrying a paternally inherited *MAU2* frameshift deletion. One sister presented with a CdLS score of 13, consistent with a classic phenotype, whereas the other scored 10. This difference likely reflects milder or absent facial dysmorphism in the younger sibling. Notably, neither sister exhibited developmental delay or intellectual disability, highlighting that cognitive deficits in *MAU2*-related CdLS are milder than in *NIPBL*-related cases. These observations raise the question of whether developmental delay and intellectual disability might be underpowered in the current CdLS scoring system, as their combined contribution accounts for only a single point of the total score.

Animal models have been instrumental in elucidating the developmental mechanisms underlying CdLS. In mice, *Nipbl* haploinsufficiency (*Nipbl*[+/−]) produces growth restriction, craniofacial anomalies, and neurodevelopmental defects that closely parallel human CdLS phenotypes[33,34]. Complete systemic loss of either *Nipbl* or *Mau2* results in embryonic lethality before E9.5, underscoring the essential function of the cohesin loader complex in early development. To circumvent this limitation, Smith and colleagues generated conditional alleles[21]. Neural crest–specific inactivation of *Mau2* resulted in severe craniofacial hypoplasia, in some cases more pronounced than with *Nipbl* deletion. Notably, double mutants (*Nipbl*[−/−]; *Mau2*[−/−]) showed milder phenotypes than *Mau2*[−/−] single mutants[21], suggesting that NIPBL may exert deleterious transcriptional activity in the absence of its binding partner. These findings imply that MAU2 not only facilitates cohesin loading but also modulates transcriptional regulatory functions. However, the developmental consequences of *MAU2* constitutional haploinsufficiency remained so far poorly understood. Although in a previous study *Mau2*[+/−] adults were indistinguishable from wild-type littermates[21], our heterozygous *Mau2* knockout mouse model recapitulates core human features, including short stature and microcephaly, thereby providing direct evidence that *MAU2* haploinsufficiency disrupts mammalian growth and brain development. The discrepancy with the previous report may reflect differences in allele design and genetic background, or the lack of standardized histopathological

procedures, an issue we have previously shown to be a common cause of false-negative findings in the literature[17].

Altogether, our findings provide compelling evidence that *MAU2* should be considered a CdLS-associated gene, while also highlighting that *MAU2* dysfunction can manifest as a distinct chromatinopathy with variable expressivity. Within our cohort, 11 out of 18 individuals were confirmed as having CdLS through molecular analyses. In three individuals, the *MAU2* variants appeared to produce only milder effects, as indicated by the presence of a *MAU2*-specific episignature alone; consistent with this, none of these three individuals exhibited classic CdLS features. These observations indicate that *MAU2* variants can give rise to a continuum of phenotypes, ranging from a mild *MAU2*-related syndrome to cases with partial or full overlap with CdLS, with corresponding differences in methylation profiles. Together, these data support the concept of a *MAU2*-related chromatinopathy characterized by variable expressivity, encompassing both a distinct syndrome and phenotypes overlapping with CdLS.

Importantly, two variants tested negative across all molecular assays (p.(Cys394Tyr), and p.(Leu528Phe)) and the corresponding individuals exhibited low CdLS clinical scores (4 points each). These findings raise two possible interpretations: either these two variants are benign and unrelated to the clinical manifestations, or they may contribute to disease through a distinct pathogenic mechanism that leads to an alternative phenotypic outcome. The shared structural eye anomalies in Individuals 3 and 5−features uncommon in CdLS−supports the latter. Additional evidence, including de novo occurrence and strong in silico predictions, further supports the pathogenicity of these variants. These findings suggest that these two variants might be associated with an alternative pathogenic mechanism. We hypothesize that they may disrupt MAU2's role in transcriptional regulation. Our previous work has shown that MAU2 is not merely a chaperone for NIPBL but also plays an active role in regulating gene expression[11], possibly by facilitating interactions with other transcription factors. It is therefore plausible that these two non-CdLS-associated variants impair MAU2's ability to interact with a specific transcriptional partner. Investigating this potential mechanism could not only broaden our understanding of MAU2's involvement in disease beyond classical CdLS, but also provide valuable insight into its poorly characterized role in transcriptional regulation.

## Methods
### *MAU2* cohort
Exome sequencing, genome sequencing, or gene panel testing were conducted at the respective referring institutions as part of routine clinical diagnostics and under approval of the relevant local institutional review boards or ethics committees. The authors did not have access to raw sequencing data; only anonymized variant and clinical information was shared for research purposes in accordance with local regulations and institutional approvals at the contributing centers.

The present study was approved by the Ethics Committee of the University Hospital Essen (UME-ID-12571). All procedures adhered to the ethical standards of the responsible institutional and/or national research committees and were conducted in accordance with the Declaration of Helsinki and its subsequent amendments.

Written informed consent for participation in the research study and for publication of anonymized data was obtained from all individuals or their legal guardians. Additional consent was secured for the publication of photographs. Participants did not receive financial compensation.

The patient cohort was collected with the assistance of GeneMatcher[35] and international collaborations. Four patients were referred with a clinical diagnosis of CdLS or a CdLS-like phenotype, while the remaining individuals were referred for various indications, including developmental delay, intellectual disability, short stature, microcephaly, or combinations thereof. Our cohort of individuals with

*MAU2* variants comprises nine males and nine females, ranging in age from 20 months to 39 years (median age 12 years). Sex was recorded for all participants based on clinical records provided by the referring institutions. As *MAU2* is located on an autosomal chromosome and there is no evidence of sex-specific effects, sex was not included as a biological variable in statistical analyses. No sex-stratified analyses were performed due to limited sample size and absence of a priori evidence for sex-dependent differences.

### *MAU2* variants and in silico predictions

Variants were mapped on the *MAU2* NM_015329.4 RefSeq transcript following HGVS recommendations[36] and classified according to ACMG Guidelines[37].

The pathogenicity of missense variants was assessed using a combination of multiple in silico prediction algorithms. The Combined Annotation-Dependent Depletion (CADD) score[38] was calculated for all variants based on GRCh37-v1.6 genomic coordinates. Evolutionary conservation of the affected MAU2 amino acids was analyzed using Alamut Visual Plus™ v1.11 (SOPHiA GENETICS, Lausanne, Switzerland). Additional prediction tools used included PolyPhen-2 (v2.2.3), REVEL (v2021-05-03), SIFT (v6.2.0), MutationTaster (v2021), and AlphaMissense[39].

### *MAU2* episignature discovery

Genome-wide DNA methylation profiling was performed using the Illumina Infinium Methylation EPIC Bead Chip, following the manufacturer's instructions (Illumina, San Diego, CA, USA). The procedures for DNA methylation analysis and episignature discovery adhered to previously established protocols[16,40]. Intensity data were processed in R (version 4.4.2) using the sesame package for normalization and background correction[41]. Probes with detection $P > 0.01$, located on sex chromosomes, near CpG interrogation sites or single nucleotide extension sites, containing single nucleotide variations, or showing cross-reactivity with other chromosomal locations other than their target regions were removed, as were arrays with >5% probe failure or those exhibiting batch effect. Genome-wide methylation density was assessed to confirm bimodal distribution, with nonconforming samples removed. Principal Component Analysis (PCA) was then used to evaluate batch structure and identify outliers.

A control cohort was randomly selected from the EpiSign Knowledge Database (EKD) and matched to *MAU2* variant carriers according to age, sex, and array type using the MatchIt package (version 4.7.0). To establish an optimal sample size, several rounds of matching were conducted. After each iteration, PCA was performed on the case and matched control samples, and any outlier controls were subsequently removed. This process was repeated until no further outliers were detected within the first two components of the PCA.

Methylation at each probe was expressed as a β-value (0–1), computed by dividing the methylated signal intensity by the sum of methylated and unmethylated signal intensities. β-values were then logit-transformed into M-values ($\log2(\beta/(1−\beta))$) to ensure homoscedasticity for downstream linear modeling. Differentially methylated probes (DMPs) were identified using multivariate linear regression in the limma package (version 3.62.2), adjusting for estimated blood cell composition[42]. Resulting *P*-values were moderated with eBayes and corrected for multiple testing using the Benjamini–Hochberg (BH) method. Probe selection was performed in three stages. First, each probe's absolute methylation difference between cases and controls was multiplied by the negative logarithm of its adjusted *P*-value, and the top-ranking probes were retained. Next, from this set, the probes with the highest area under the curve (AUC) values were selected based on receiver operating curve (ROC) characteristic analysis. Lastly, probes showing pairwise Pearson correlation coefficients greater than 0.7 between cases and controls were removed to improve genome-wide representation, resulting in the final probe set. These probes were then analyzed by hierarchical clustering using Ward's method on Euclidean distances and multidimensional scaling (MDS) to evaluate separation between cases and controls.

A binary support vector machine (SVM) classification model was developed using the e1071 R package (version 1.7.16), according to previously described procedures[16,40]. The classifier generates methylation variant pathogenicity (MVP) scores (0–1), which provide a measure of prediction confidence for each disorder: scores close to 1 indicate a high probability that the sample's methylation pattern corresponds to the target syndrome, while scores near 0 indicate a methylation pattern characteristic of controls. The model was trained using *MAU2* variant carriers with their matched controls, as well as 75% of other controls and 75% of samples from other rare disorders obtained from the EKD. The model was tested using the remaining 25% of control samples together with the remaining 25% of samples from other rare disorders within the EKD. Specificity was evaluated using the proportion of samples in the "Unresolved" column and an MVP threshold of 0.25.

Robustness was assessed by multiple rounds of leave-one-out cross-validation (LOOCV), where in each round one *MAU2* sample was withheld for testing while the remaining *MAU2* samples guided probe selection and model construction. The resulting model was then applied to classify the withheld sample. Results were visualized using heatmaps, MDS plots, and MVP plots to illustrate sample classification and the distinction between the test sample and the other samples in each round.

To evaluate whether *MAU2* samples align with the known CdLS episignature, DNA methylation data were analyzed using the clinically validated EpiSign classifier, following previously established methods[16,27–29]. Normalized and background-corrected β-values derived from the EPIC V2 arrays were processed through the established SVM model, which references the EKD of disorder-specific and control profiles to generate MVP scores (0–1). MVP scores near 1 indicate a high probability that the sample matches the CdLS signature. A positive classification is typically defined by an MVP score greater than 0.5. The final matched EpiSign result is generated using these scores, along with the assessment of hierarchical clustering and multidimensional scaling.

### Mammalian two-hybrid assay

A fragment of NIPBL containing amino acids 1–300 was inserted into the pCMV-BD expression plasmid (#211342, Agilent Technologies, Santa Clara, CA, USA). The full-length open reading frame of MAU2 was cloned into the pCMV-AD plasmid (#211343, Agilent Technologies). MAU2 mutant constructs were generated through site-directed mutagenesis based on divergent PCR in combination with 5'-phosphorylated primers. Constructs harboring the following variants were generated for this assay: p.(Cys50Ser), p.(Val54_Pro56del), p.(Ala309_Lys322del), p.(Gln310_Ala316del), p.(Lys322del), p.(Ala378_Gln380delinsAsp), p.(Leu381Pro), p.(Cys394Tyr), p.(Leu528Phe), p.(Asp534His), as well as the synonymous variant, frequent in the healthy population, p.(Thr355Thr). HEK293 cells were transiently transfected in 24-well plates with FuGene-HD transfection reagent (E2311, Promega, Madison, WI, USA), according to the manufacturer's instructions. Each well was transfected with 250 ng of the pCMV-BD-NIPBL 1-300aa, 250 ng of the pCMV-AD-MAU2 wild-type or mutant constructs, 250 ng of the Firefly Luciferase reporter plasmid and 2.5 ng of the phRG-TK Renilla luciferase expression plasmid. Activity of Firefly and Renilla luciferases was measured 24 h post-transfection with the Dual Luciferase Reporter Assay System (E1980, Promega) using the GloMax Discover System (Promega). All measurements were performed in triplicate in at least six independent experiments. Relative luciferase activity, indicating the strength of the interaction, was determined as the triplicate average of the ratio between the Firefly and the Renilla luciferase activity.

The statistical significance of the mammalian two-hybrid results was first assessed using the Kruskal-Wallis test to evaluate differences across groups. After identification of a significant effect, pairwise comparisons were performed using two-sided Dunn's test, with a Bonferroni correction applied to account for multiple comparisons.

## Expression controls of mammalian two-hybrid constructs

Expression of exogenous MAU2 wild type and mutant constructs (pCMV-AD-MAU2) was verified by Western Blot. HEK293 cells were transiently transfected in 10 cm plates with the jetOPTIMUS® DNA Transfection Reagent (101000006, Polyplus, Illkirch, France), following the manufacturer's instructions. Each plate was transfected with 10 µg of the corresponding MAU2 construct. Cells were harvested 24 h after transfection. After resuspension in 250 µl of lysis buffer (150 mM NaCl, 20 mM NaF, 50 mM Tris pH 7.5, 0.1% NP40, 1 mM EGTA, protease inhibitor cocktail (04693132001, Merck, Darmstadt, Germany)), samples were incubated on ice for 60 min. Protein lysates were obtained by high-speed centrifugation at 4 °C for 30 min. Protein concentration was determined using the BCA Protein Assay Kit (#23225, Pierce, Rockford, IL, USA) following the manufacturer's protocol. Protein expression was analyzed by Western blotting using Mini-PROTEAN® TGX™ gels (4568106, Bio-Rad, Hercules, CA, USA). Briefly, 20 µg of each protein lysate were separated by SDS-PAGE in 1X Tris-Glycine-SDS running buffer and transferred to a nitrocellulose membrane using a semi-dry transfer system (Trans-Blot® SD Semi-Dry Transfer Cell, Bio-Rad) at 25 V for 30 min, following the manufacturer's protocol. After transfer, the membrane was blocked with EveryBlot Blocking Buffer (#12010020, Bio-Rad) for 15 min at room temperature.

Membranes were subsequently incubated overnight at 4 °C with the following primary antibodies: MAU2 (1:1000; ab183033, Abcam, Cambridge, MA, USA) and NF-κB (1:500; sc-372, Santa Cruz Biotechnology, Dallas, TX, USA). Antibodies were diluted in 5% bovine serum albumin (BSA) in Tris-buffered saline with 0.1% Tween-20 (TBST). Following incubation, the membrane was washed four times with TBST, each wash lasting ten minutes. It was then incubated for 1 h at room temperature with the appropriate HRP-conjugated secondary antibody, diluted in the same buffer used for the primary antibodies: goat-α-rabbit (#31460, Thermo Fisher Scientific, Waltham, MA, USA) diluted 1:10,000 for the MAU2- and NF-κB membranes. Following four washes with TBST and two washes with TBS, protein bands were detected using an enhanced chemiluminescence (ECL) substrate (SuperSignal West Femto ECL, #34095, Thermo Fisher Scientific) and visualized with the Intas ChemoStar Touch imaging system (Intas Science Imaging Instruments, Göttingen, Germany).

## Cyclohexamide chase assay

For the cycloheximide (CHX) chase assay, HEK293 cells expressing pCMV-AD-MAU2 constructs were treated 24 h post-transfection with CHX at a final concentration of 100 µM for 8 h. Following treatment, cells were harvested, lysed, and subjected to western blot analysis (against NF-κB) as described in the previous paragraph. For each sample, the signal corresponding to exogenous MAU2-NF-κB was normalized to endogenous NF-κB. The resulting ratio for each mutant was then normalized to the corresponding ratio obtained for the treated wild-type sample. Six independent lysates were analyzed, with four independent western blots performed per lysate. Technical replicates were averaged to generate a single value per biological replicate, which was used for statistical analysis. Statistical significance was assessed using two-sided Mann–Whitney U test with Bonferroni correction for multiple comparisons. The magnitude of the effect was estimated independently as the median fold change in normalized MAU2-NF-κB signal relative to wild-type. Effect sizes were classified a priori into descriptive categories: large decrease (median ratio <0.3),

moderate decrease (0.3–0.6), small decrease (0.6–0.8), no change (0.8–1.2), small increase (1.2–1.5), moderate increase (>1.5).

## RNA isolation, cDNA synthesis and Real-Time PCR

RNA extraction was carried out with the ReliaPrep™ RNA Cell Miniprep System (Promega) according to the manufacturer's instructions. The LunaScript® RT SuperMix Kit (E3010L, New England Biolabs, Ipswich, MA, USA) was used to retro-transcribe 1 µg of RNA. cDNA synthesis was performed in two independent reactions for each of the three biological replicates of the same sample.

The expression levels of the transcripts of interest were assessed using the qPCRBIO Probe Mix Hi-ROX assay for Real-Time PCR (PB20.23-05, PCR Biosystems, London, UK). The investigation was run on the LightCycler 480 (Roche, Basel, Switzerland). The following TaqMan gene expression assays were used for the analysis: Hs01062386_m1 (for the *MAU2* transcript) and Hs00209846_m1 (for the *NIPBL* transcript) (Thermo Fisher Scientific). The *SNAPIN* and *NADH* genes were selected as endogenous normalizer and amplified with the TaqMan gene expression assay ID Hs00276176_m1 and Hs00190020_m1 (Thermo Fisher Scientific). Relative gene expression was determined using the ΔΔCt method[43]. Two-sided Mann–Whitney U test, followed by Bonferroni correction for multiple comparisons, were used to assess the statistical significance of the qPCR results.

## Western Blot on fibroblasts of affected individuals

Fibroblast pellets were resuspended in 100 µl of RIPA lysis buffer (#89900, Thermo Fisher Scientific) supplemented with protease inhibitor cocktail (04693132001, Merck). Samples were incubated on ice for 30 min. Protein lysates were then obtained by high-speed centrifugation at 4 °C for 30 min. Western Blot experiments were carried out as previously described in the "Expression Controls for mammalian two-hybrid constructs" section unless otherwise specified. The following primary antibodies were used for detection: MAU2 (1:1000; ab183033, Abcam), Actinin (1:1000; sc-17829, Santa Cruz Biotechnology), and NIPBL (1:500; KT54, Absea). The secondary antibodies were diluted as follows: goat-α-rabbit for MAU2 (1:10,000; #31460, Thermo Fisher Scientific), goat-α-mouse for Actinin (1:100,000; #31430, Thermo Fisher Scientific), and goat-α-rat for NIPBL (1:10,000; #A10549, Thermo Fisher Scientific). An uncropped and unprocessed scan of the corresponding Western blot can be found in the Source Data file.

## Mau2^tm1b(KOMP)Wtsi knockout mouse

The care and use of mice in the study was carried out in accordance with UK Home Office regulations, UK Animals (Scientific Procedures) Act of 1986 under 3 UK Home Office licenses that approved this work, which were reviewed regularly by the local Animal Welfare and Ethical Review Body. The ethical approval number for studies on animals was granted under two UK Home Office project licenses (80/2485 and P77453634). These licences were subject to regular review by the Wellcome Sanger Institute Animal Welfare and Ethical Review Body. The *Mau2^tm1b(KOMP)Wtsi* mouse was generated at the Wellcome Sanger Institute by homologous recombination in embryonic stem cells using the knockout-first allele method[20]. In this strategy, an exon common to all transcripts (exon 5) was targeted, upstream of which a LacZ cassette was inserted. Exon 5 of the *Mau2* allele, flanked bilaterally by *loxP* sequences, was deleted using Cre recombinase that recognizes *loxP* sites, thereby producing the *Mau2^tm1b(KOMP)Wtsi* knockout allele. The strain used in this study *is Mus musculus domesticus* and the mice were bred on a pure C57BL/6 N inbred genetic background. A core colony was established using mice obtained from Taconic Biosciences. To minimize genetic drift, the colony nucleus was regularly refreshed (approximately every 10 generations) and cryopreserved. All animals used in the experiments were 16 weeks of age. Mice were given a breeders chow (Mouse Breeder Diet 5021, 9% crude fat content, 21%

kcal as fat, 0.276 ppm cholesterol, Labdiet, London, UK) from weaning. All mice were given water and diet ad libitum. Mice were maintained in a specific pathogen free unit with sentinel mouse monitoring on a 12 h light/12 h dark cycle with lights off at 7:30 pm and no twilight period. The ambient temperature was $21 \pm 2^{o}C$ and the humidity was $55 \pm 10\%$. All animals were regularly monitored for health and welfare concerns and were additionally checked prior to and after procedures. Mice were typically housed using a stocking density of 3−5 mice per cage (overall dimensions of caging: (LxWxH) 365 x 207 × 140 mm, floor area 530 cm²) in individually ventilated caging (Tecniplast Seal Safe1284L) receiving 60 air changes per hour. In addition to Aspen bedding substrate, standard environmental enrichment of 1−2 nestlets, a cardboard tunnel and, three wooden chew blocks were provided.

Through collaboration with the Sanger Institute Mouse Genetics Project, we obtained weight and body height measurements of seven *Mau2*^+/- female mice and 340 baseline WT female mice, as well as seven *Mau2*^+/- male mice and 345 baseline WT male mice. We also received brain samples from *Mau2*^+/- mice and performed comprehensive neuroanatomical studies on three *Mau2*^+/- mutants and 44 baseline WT mice at 16 weeks of age, as previously described[22,23]. A validated statistical model (G*Power) was applied to detect neuroanatomical defects with an effect size ≥10% at 80% power[17]. Brains were fixed in 4% buffered formalin for 48 h, transferred to 70% ethanol, and paraffin-embedded. Coronal sections (5 µm thickness) were obtained at Bregma +0.98 mm and Bregma −1.34 mm (Allen Mouse Brain Atlas) using a Leica RM 2145 microtome. Sections were double-stained with 0.1% Luxol Fast Blue (Solvent Blue 38; Sigma-Aldrich) and 0.1% Cresyl violet acetate (Sigma-Aldrich), then scanned at 20× resolution using the Nanozoomer whole-slide scanner 2.0HT C9600 series (Hamamatsu Photonics, Japan). Images were quality controlled for sectioning accuracy, asymmetries, and histological artifacts. All samples were assessed for cellular ectopia. 63 brain parameters (comprising left and right hemispheres) were measured blind to the genotype across two coronal sections as described in Mikhaleva et al.[22]. Data were analyzed using a two-tailed Student's t-test assuming equal variances to determine whether a brain region was associated with neuroanatomical defect or not. All animal studies were conducted in accordance with French regulations (EU Directive 86/609 – French Act Rural Code R 214-87 to 126) and all procedures were approved by the local ethics committee and the Research Ministry.

### Use of AI tools in data analysis and manuscript preparation
ChatGPT (OpenAI, San Francisco, CA, USA) was used to assist in data analysis (R scripting) and to provide language editing suggestions for this manuscript. The authors reviewed the output for accuracy.

### Reporting summary
Further information on research design is available in the Nature Portfolio Reporting Summary linked to this article.

## Data availability
All variants have been submitted to the ClinVar Database. Accession numbers are SCV006580334 - SCV006580348. Additional data and materials supporting the findings of this study, including in vitro experimental data, cell lines and constructs, are available from the corresponding author upon request. The deposition of individual epigenomic or any other personally identifiable data that has not previously been made publicly available for samples in the EpiSign Knowledge Database (EKD) is prohibited from deposition in publicly accessible databases due to institutional and ethical restrictions. Specifically, these include data and samples submitted from external institutions to the EKD that are subject to institutional material and data transfer agreements, data submitted to London Health Sciences for episignature assessment under Research Services Agreements, and research study cohorts under Institutional Research Ethics Approval

(Western University REB 106302 and REB 116108). Source data are provided in this paper.

## Code availability
No custom code was generated for this study. Publicly available software packages used are described and cited in the Materials and Methods section. EpiSign is a commercial software tool and is not publicly available.

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

## Acknowledgements

We are deeply grateful to the individuals of our cohort and their families for their trust, willingness to share clinical data, and generous contribution of time and samples, without which this study would not have been possible. This study made use of data generated by the International Mouse Phenotyping Consortium (IMPC, www.mousephenotype.org). We gratefully acknowledge the IMPC and its member institutions for their efforts in generating and providing access to phenotyping data. In particular, we are extremely grateful to Valérie E. Vancollie and Christopher J. Leliott, as well as members of the Sanger Institute Mouse Pipelines teams and the Research Support Facility, for the production and management of the *Mau2+/-* mouse line used in this study. We thank Dorinda Wright, Susan Mary Smith, and Lindsey Proctor for histological work at HistologiX Ltd, and the students and technicians of the Yalcin's laboratory for their contributions to mouse phenotyping, in particular Kevin Navarro, Helen Whitley, Christel Wagner, Marie-Christine Fischer, Anna Mikhaleva, Rebecca Balz, and Léo Gagliardi. I.P. is supported by the CHOPS Rare Disease Foundation, and F.J.K. receives support from the Dr. Holger Müller Stiftung. S.C.C. is a Senior Lecturer at the University Bourgogne-Europe. B.Y. is an INSERM investigator supported by grants from the Agence Nationale de la Recherche and the IFCPAR/CEFIPRA (grant no. 6503-J). A.A.L.J. is supported by the São Paulo Research Foundation (FAPESP; 2022/10107-6), and the National Council for Scientific and Technological Development (CNPq; 303294/2020-5). F.J.R., J.P., B.P., and M.G.S. are investigators supported by grants from the Instituto de Salud Carlos III (Grant Ref. PI23-01370) and Diputación General de Aragón-FEDER: European Social Fund [Reference Group B32_20R]. A.K., F.J.K. (Essen, Germany), and B.Y. (Dijon, France) are members of the European Reference Network (ERN) ITHACA. P.M.B. was supported by award K08NS117891 from the U.S. National Institute of Neurological Disorders and Stroke (NINDS) and an award from the Boston Children's Hospital Office of Faculty Development. T.O. was supported by AMED under grant number JP23ek0109678.

## Author contributions

Conceptualization: I.P., F.J.K. Investigation: I.P., A.H., M.G.S., S.C.C., L.D., D.K., J.B., J.K., L.S., J.E., R.K., B.Y., and K.S.W. Formal analysis: I.P., A.H., M.G.S., L.D., S.C.C., J.K., E.L., B.S., B.Y., and K.S.W. Resources: J.W., P.M.B, E.M.K., L.A., H.M.A., F.S.A., M.J.A., S.E.A., D.B., K.C., L.C., A.R.C., A.A.L.J., A.C.L., B.D., M.O.H., P.J., A.O.K., K.K., G.J.L., E.M., J.M.C., L.K.K., S.M., G.N., Y.N., T.O., J.Pi., J.Pr., B.P., F.J.R., E.Ra., C.R., E.Ru., S.Sa., H.E.S., S.St., E.A.S., S.H.E., and A.K. Visualization: I.P., A.H., M.G.S., and L.D. Writing—original draft: I.P., A.H., B.Y., K.S.W., and F.J.K. Writing—review and editing: All authors reviewed and edited the manuscript.

## Funding

## Competing interests

A.A.L.J. reported receiving consulting fees and grants from BioMarin, Novo Nordisk and BridgeBio. S.E.A. is a co-founder and CEO of Medigenome, Swiss Institute of Genomic Medicine. F.S.A is a paid employee of Lifera Omics. A.R.C is a former employee and current shareholder of Illumina. All other authors declare no conflict of interest.

## Additional information

Ilaria Parenti [1] ✉, Alina Hesters [1], Marta Gil-Salvador [2], Laura Duffy [3,4], Deniz Kanber[1], Jasmin Beygo[1], Jennifer Kerkhof [3], Laura Steenpaß[5,6], Elsa Leitão [1], Julia Woestefeld [1], Philip M. Boone[7,8], Emeline M. Kao[9], Lama Alabdi[10], Hesham M. Aldhalaan[11], Fowzan S. Alkuraya[10,12,13], Muneera J. Alshammari[14], Stylianos E. Antonarakis [15], Donald Basel[16], Kevin Cassinari[17], Laurana de Polli Cellin[18], Amanda R. Clause [19,20], Alexander Augusto de Lima Jorge [18], Andréa de Castro Leal [21], Stephan C. Collins [22,23], Benjamin Durand [24], Juliane Eckhold[1], Mais O. Hashem[10], Parul Jayakar[25], Arif O. Khan[26,27], Kohji Kato[28,29,30], Regina Kubica[1], Gholson J. Lyon [31,32,33], Elaine Marchi[31,32], Julie McCarrier[16], Lara K. Kimmig[1], Seiji Mizuno[34], Gael Nicolas [17], Yosuke Nishio [29,30,34,35], Tomoo Ogi [30], Juan Pié[2], Jordyn Prell[16], Beatriz Puisac [2], Feliciano J. Ramos [2,36], Emmanuelle Ranza[15], Claire Redin[15], Eric Rush[37,38], Shinji Saitoh [39], Hanan E. Shamseldin[10], Susan Starling[37], Esteban Astiazaran-Symonds[40], Sara H. Eltahir[14], Alma Kuechler[1], Bekim Sadikovic [3,4], Binnaz Yalcin [22,23], Kerstin S. Wendt [41] & Frank J. Kaiser[1,42]

[1]Institute of Human Genetics, University Hospital Essen, University of Duisburg-Essen, Essen, Germany. [2]Unit of Clinical Genetics and Functional Genomics, Department of Pharmacology and Physiology, School of Medicine, University of Zaragoza, CIBERER and IIS-Aragon, Zaragoza, Spain. [3]Verspeeten Clinical Genome Centre, London Health Science Centre, London, ON, Canada. [4]Department of Pathology and Laboratory Medicine, Western University, London, ON, Canada. [5]Human and Animal Cell Lines, Leibniz-Institute DSMZ-German Collection of Microorganisms and Cell Cultures GmbH, Braunschweig, Germany. [6]Zoological Institute, Technische Universität Braunschweig, Braunschweig, Germany. [7]Division of Genetics and Genomics, Boston Children's Hospital, Boston, MA, USA. [8]Center for Genomic Medicine, Massachusetts General Hospital, Boston, MA, USA. [9]Institutional Centers for Clinical and Translational Research, Boston Children's Hospital, Boston, MA, USA. [10]Department of Translational Genomics (Genomic Medicine Centre of Excellence (GMCoE)), King Faisal Specialist Hospital and Research Center, Riyadh, Saudi Arabia. [11]Autism Department King Faisal Specialist Hospital and Research Center, Riyadh, Saudi Arabia. [12]Lifera Omics, Riyadh, Saudi Arabia. [13]Department of Anatomy and Cell Biology, College of Medicine, Alfaisal University, Riyadh, Saudi Arabia. [14]Department of Pediatrics, King Saud University Medical City, King Saud University, Riyadh, Saudi Arabia. [15]Medigenome, Swiss Institute of Genomic Medicine, Geneva, Switzerland. [16]Division of Genetics, Department of Pediatrics, Medical College of Wisconsin, Milwaukee, WI, USA. [17]Department of Genetics and Reference Center for Developmental Disorders, Univ Rouen Normandie, Normandie Univ, Inserm U1245 and CHU Rouen, Rouen, France. [18]Genetic Endocrinology Unit, Endocrinology Division, Hospital das Clínicas da Faculdade de Medicina da Universidade de São Paulo (HC-FMUSP), São Paulo, SP, Brazil. [19]Illumina Laboratory Services, Illumina Inc., San Diego, CA, USA. [20]Neurology, Washington University in St. Louis, St. Louis, MO, USA. [21]Department of Integrated Health, State University of Para, Santarem, Brazil. [22]Université Bourgogne-Europe, Inserm U1231, Dijon, France. [23]Institut NeuroMyoGène, Unité Physiopathologie et Génétique du Neurone et du Muscle, Université Claude Bernard Lyon 1 CNRS UMR 5261, Inserm U1315, Lyon, France. [24]Service de Génétique Médicale, Institut de Génétique Médicale d'Alsace (IGMA), Hôpitaux Universitaires de Strasbourg, Strasbourg, France. [25]Division of Genetics & Metabolism, Nicklaus Children's Hospital, Miami, FL, USA. [26]Ophthalmology, Integrated Surgical Institute, Cleveland Clinic Abu Dhabi, Abu Dhabi, UAE. [27]Department of Ophthalmology, Cleveland Clinic Lerner College of Medicine of Case Western Reserve University, Cleveland, OH, USA. [28]School of Biochemistry, Faculty of Life Sciences, University of Bristol, Bristol, UK. [29]Department of Pediatrics, Nagoya University Graduate School of Medicine, Nagoya, Japan. [30]Department of Genetics, Research Institute of Environmental Medicine, Nagoya University, Nagoya, Japan. [31]Department of Human Genetics, NYS Institute for Basic Research in Developmental Disabilities, Staten Island, NY, USA. [32]George A. Jervis Clinic, NYS Institute for Basic Research in Developmental Disabilities, Staten Island, NY, USA. [33]Biology PhD Program, The Graduate Center, The City University of New York, New York, NY, USA. [34]Department of Pediatrics, Aichi Developmental Disability Center, Kasugai, Japan. [35]Medical Genomics Center, Nagoya University Hospital, Nagoya, Japan. [36]Clinical Genetics Unit, Service of Pediatrics, University Hospital 'Lozano Blesa', University of Zaragoza School of Medicine, Zaragoza, Spain. [37]Division of Clinical Genetics, Children's Mercy Kansas City, University of Missouri Kansas City School of Medicine, Kansas City, MO, USA. [38]Department of Internal Medicine, University of Kansas School of Medicine, Kansas City, MO, USA. [39]Department of Pediatrics and Neonatology, Nagoya City University Graduate School of Medical Sciences and Medical School, Nagoya, Japan. [40]Department of Medicine, College of Medicine-Tucson, University of Arizona, Tucson, AZ, USA. [41]Erasmus Medical Centre, Department of Developmental Biology, Rotterdam, The Netherlands. [42]Essener Zentrum für Seltene Erkrankungen (EZSE), University Hospital Essen, Essen, Germany. ✉e-mail: ilaria.parenti@uk-essen.de

