## [Transparent Peer Review file · Nature Communications]

Pathogenic variants in the cohesin loader subunit MAU2 underlie a distinct Cornelia de Lange Syndrome subtype

Corresponding Author: Dr Ilaria Parenti

Version 0:

Reviewer comments:

Reviewer #1

(Remarks to the Author)

This work by Parenti et al. is a very important well-written article describing individuals with MAU2 variants, previously minimally reported and likely with many variants lethal. They make convincing arguments that MAU2 be considered to be a new CdLS-causal gene in that certain variants found in MAU2 cause CdLS as supported both clinically and by epigenetic signature; this is also supported by findings in the MAU2 knockout mouse. The detailed mouse description is excellent, and brain findings very interesting and novel. The mammalian two-hybrid results also are supportive of MAU2's complementary role with NIPBL. MAU2 being a CdLS-causal gene has long been suspected, so this article will be well received.

The patients with the variants are thoroughly described, including using the international consensus clinical diagnostic criteria for CdLS. It is somewhat less clearly stated that those patients with lower clinical diagnostic criteria scores or less typical findings as seen in Table 1 do not resemble patients with CdLS, including in Figure 6. (It is also unfortunate that most of the patients' photos are not available for the manuscript.) It would be helpful if the authors could describe a non-CdLS MAU2 facial phenotype or if they feel that it is nonspecific. In addition, patient 5 (non CdLS) appears to have a repaired cleft lip, but only cleft palate is mentioned in the supplemental table. Clefting is not mentioned otherwise in Table 1, and typically only cleft palate is seen in patients with CdLS. I wondered if any of the other patients (especially those without the CdLS higher clinical scores or CdLS episinature) had cleft lip, and if so, could that be associated with MAU2 (or if this was the only one and thus likely unrelated)? It would be helpful to mention that hearing loss, seen so commonly in CdLS, was not seen in most patients with the MAU2 variants, since that may set it apart. Finally, in the abstract and elsewhere, microcephaly and short stature are described as being consistent features, but really are not present in 20-30% of the patients according to Table 1; perhaps the word "consistent" should be replaced with "major" or "in the majority" or something less universal?

Otherwise, no further comments, and strongly recommend publication.

Reviewer #2

(Remarks to the Author)

Cornelia de Lange syndrome (CdLS) is a neurodevelopmental disorder of genetic origin that is characterised by typical features including craniofacial dysmorphism, hirsutism, growth and cognitive delays, as well as system impairment such as gastrointestinal tractus and heart. Of note, clinical manifestations are of variable severity amongst patient population, and while typical, severe CdLS cases are easily diagnosed, diagnosis for the lower end of the spectrum often needs to discriminate between CdLS and other related neurodevelopmental syndromes. In that aim, an international effort has established score-based guidelines to provide clinicians with common diagnosis practices.

CdLS arises from heterozygous pathogenic variants in a number of genes encoding members of cohesin complex and of its regulators. Vast majority of cases (up to 70 %) involve NIPBL gene, which encodes the protein NIPBL that, together with its binding partner MAU2, forms the Kollerin complex and loads cohesin complexes onto chromatin. Importantly, Kollerin also acts in the regulation of gene expression independently of cohesin. 15 % of remaining cases are accounted for by variants in cohesin-related genes (SMC3, SMC1A, HDAC8, RAD21 and BRD4) and are characterised by broad clinical manifestations and, in some cases, atypical presentations.

So far, only two patients with variants in the NIPBL partner MAU2 have been reported, including one by the authors. Pathophysiological mechanisms for MAU2 variants are lacking due to the small number of cases, albeit impaired

NIPBL/MAU2 interaction and protein folding and stability have been proposed by the same authors. In addition, and for the same reason, it is unclear whether MAU2 variants result in classical CdLS presentation or constitute a distinct subtype of CdLS.

In this new manuscript entitled "Pathogenic variants in the cohesin loader subunit MAU2 lead to a new Cornelia de Lange Syndrome subtype", Parenti and colleagues report eighteen patients carrying different types of variants in MAU2, which they used to interrogate common and distinct clinical features and disease-relevant molecular mechanism.

Once having presented the 15 distinct MAU2 variants they identified and that span its entire coding sequence, the authors first assessed patient DNA methylation profiles (episignatures) aiming at defining or refining CdLS episignatures. Out of 14 cases studied, 10 of them showed a Episign methylation profile similar to that previously established for CdLS patients. As four cases did not match CdLS episignature, the authors tested the possibility of specific MAU2 episignature(s) and were able to establish two distinct MAU2 Episignatures, which both allow clustering of most MAU2 patients, though with some discordances.

As an altered physical interaction between NIPBL and MAU2 impairs NIPBL folding and stability, and therefore its transcriptional activity, the authors next addressed the impact of the ten MAU2 substitution variants in mammalian two-hybrid assays. These experiments revealed that 6 out of the 10 tested variants did impact negatively on the ability of NIPBL/MAU2 to activate transcription of reporter gene (luciferase). In addition, the authors assessed to which extent MAU2 variants impacted on MAU2 stability as a cause for reduced luciferase expression by probing MAU2 protein levels produced in these assays by immunoblots, which did not allow to address this particular point unambiguously.

Next, the authors assessed the consequence of truncating, out-of-frame variants on the amounts of NIPBL and MAU2 at protein and mRNA levels using patient-derived dermal fibroblasts obtained from one patient and her father from whom variant was inherited. RT-qPCR experiments revealed that MAU2 mRNA levels are reduced as compared to control individuals while the situation for NIPBL is unclear. Immunoblotting analysis of MAU2 and NIPBL in the same cell samples suggested to some extent a reduction of both protein levels in the patient's cells and her father's.

With a unique cohort of 18 "MAU2 patients" at their disposal, the authors then exhaustively analysed their clinical features in order to identify subsets of them that would show some specificity/peculiarity between the whole MAU2 cohort compared to the 11 MAU2 cases molecularly assigned to CdLS. This comparison highlighted a slightly higher frequency in the CdLS MAU2 patients as compared to the whole cohort of a few facial features while frequency in all other features were similar between the two groups of MAU2 patients.

Finally, the authors present the characterisation of a Mau2 heterozygous null mouse, which exhibits shorter size, microcephaly and a set of structural brain anomalies, in agreement with previously reported CdLS mouse models, and recapitulates some of CdLS hallmark features.

Overall, this manuscript is an interesting piece as it reports on the largest MAU2 patient cohort published so far, which represents useful resources for clinicians, and unambiguously establishes MAU2 variants as a cause of CdLS. Further, the authors report that MAU2 patients present specific DNA-methylation based episignatures, which are distinct from that of the established CdLS episignature and may well have utility in clinical practices. Parenti and colleagues also evidenced how these variants impact on the ability of the Kollerin complex to form, possibly hinting at pathophysiological mechanisms. This study largely focuses on the presentation and phenotypical characterisation of the MAU2 cohort, and, albeit mammalian two-hybrid assay does point toward impaired interaction between MAU2 and NIPBL, do not improve significantly molecular understanding of CdLS. Along the same line, the presented mouse model constitutes an interesting confirmation of MAU2 haploinsufficiency as a cause of murine CdLS-like features, but does bring any information as to how it results into described brain structure anomalies. Furthermore, and if most of this manuscript reads well, some parts are a bit confusing, both concerning results and text (see below for specific comments).

In conclusion, I would recommend the authors to consider publishing their work in a journal with a scope more focused in human genetics, although I would be happy to reconsider my position if the authors were able to improve significantly on the molecular mechanism at play here.

Main comments (in no particular order):

1- Regarding description of Figure 2, the authors wrote "three of these variants...., but grouped with controls when analysed separately." (l. 194-197). This is a bit unclear to me what it means and what the implications of such a result are (Sup Fig 1). This will need more detailed description.

2- Similarly confusing to me is the identification of the two MAU2 episignatures. Were they obtained from distinct clustering analyses? Were only 9 and 7 samples considered for the signature discovery? If so, why? Which MAU2 sample falls into each signature? What about non MAU2 CdLS cases (i.e., are MAU2 signatures discriminating between MAU2 and other CdLS)?

3- Regarding Figure 3: I agree with the authors that while six variants impact on detected luciferase activity, four do not; however I am not convinced by the corresponding interpretation that MAU2 episignature has much to do with the results: indeed, all tested variants that show reduced luciferase activity exhibit CdLS signature, regardless of MAU2 signatures. Similarly, all variants that do not affect luciferase are negative for the CdLS signature, irrespective of the MAU2 signature. Therefore, and unless I missed something, this assay almost perfectly identifies CdLS signature positive and negative cases, while it does not do so for MAU2 ones.

In the same set of experiments, presented in Sup Fig 4, results of protein expression analyses are not very clear: NIPBL and MAU2 seem to be challenging to blot for and, as a consequence, it is very hard to assess clearly differences in protein amounts of MAU2, NFKB and GAPDH on the presented images. Same comment applies to Flag immunoblot. Finally, the fact that the authors argue that differences in protein amount caused by different variants depend at least partly on the plasmid used for ectopic expression further confuses interpretation for the reader.

4- In Figure 4: I acknowledge authors' efforts and will of completeness as they assessed mRNA abundance expressed relative to two different normalisation genes. However, if MAU2 mRNA reduction is observed in cells from the patient and from her father independently of the normalisation gene used, the fact that it shows a reduction for NIPBL in one case and no changes in the other one is surprising, and adds difficulty to interpretate these results.

5- In Figure 5. The comparison of canonical clinical features between whole MAU2 cohort vs MAU2 patients that exhibit CdLS signature highlights some differences at the facial level, which is interesting per se. How different would it be if CdLS positive patients vs CdLS negative patients were compared? It might reveal clear differences between CdLS and MAU2. Also, how does the 18 patient MAU2 cohort compare to classic CdLS patients (e.g., carrying NIPBL variants)? Any significant differences may strengthen the reality of a MAU2-specific subgroup of CdLS.

Minor comments

- 1- In Figure 1, degrees of variation intolerance, as stated in the legend, are represented by different colours. I guess these degrees are also represented by heights of the curves at the different positions. If so; please indicate in the legend, and if not, please precise.
- 2- As identification of MAU2 epesignatures represent a main result of the presented work, I feel like Sup Fig 2 and 3 would nicely fit in the main text (in Figure 2 or as a distinct figure).
- 3- I would replace Fig4E with SUP Fig6, as it gives much clearer indication of MAU2 protein reduction. Ideally, NIPBL would be blotted on the same experiment.

Reviewer #3

(Remarks to the Author)

Variants in *Mau2* causing a CdLS-like phenotype have been described previously (as acknowledged by the authors) – this manuscript describes the largest cohort to date and characterizes the effects these variants mechanistically and in an animal model – adding credence and support to the pathogenicity of these variants and expanding the cohesin-related genes that contribute to the causality of CdLS and related diagnoses. The manuscript is well written and structured with synergistic clinical, molecular, cellular and animal modeling to support the authors' hypotheses and claims. Parenti et al., propose (and provide convincing data) that that pathogenic variants in MAU2 result in a CdLS subtype and that MAU2 should be recognized as a new CdLS-associated gene. This work will contribute significantly to the field by expanding our understanding of key regulators of cohesin's role in contributing to mammalian structural and developmental differences and expanding our understanding how these proteins regulate developmental gene expression.

- 1) Recommend replacing the terms "disease" and "disorder" with "diagnosis"
- 2) Recommend replacing "patients" with "individuals" when describing affected people
- 3) How do the authors explain that one of their pathogenic variants was found in gnomAD (p.(Cys50Ser)) in 6 individuals? This variant in the reported individual was paternally inherited – what was the father's phenotype (in the supplementary table they list him as 165 cm tall – which plots onto the standard growth curves)? The methylation pattern for this individual also clusters with the controls and the two-hybrid assay suggests benign.
- 4) Would be ideal to include all parents who carry the variants argued to be causative in the "affected" individuals in this manuscript, although the authors state that they were unable to obtain sufficient clinical information on all of these individuals – unclear if all parents who carry these variants are truly "affected" but more information would help delineate the clinical spectrum of MAU2-related CdLS, and whether or not non-penetrance is possible (or if some of these variants may in fact be benign). Possible to include any photographs of affected/carrier parents?
- 5) Patient 5 – looks like also may have cleft lip in photo (listed as "midline cleft palate")?
- 6) Do the authors know if all individuals treated with GH were actually GH deficient on testing (they mention it for a few but for others – presume that information was not available) – if GH levels were not known in these individuals it should be stated in Suppl Table 3.
- 7) Can the authors expand on how this cohort was identified – the bias may be that they were enrolled in various CdLS studies ("international collaborators") and/or selected from individuals obtaining exome or genome sequencing (e.g. through GeneMatcher ascertainment) which would aggregate them with likely syndromic or non-syndromic developmental delay of intellectual disability that might introduce a bias in the clustering of phenotypic features around the various testing parameters undertaken as part of this study.
- 8) It would be of interest for the authors to expand a bit (beyond the one sentence mention of the double heterozygous mouse model) as to why there is significant phenotypic discrepancy between NIPBL related CdLS and MAU2-related CdLS? Given the direct interaction and functional facilitation of these two proteins one might expect a more significant overlap between the phenotypes – especially in those with haploinsufficient mutational mechanisms – but only severe pathogenic variants in CdLS cause the truly "classical" severe CdLS phenotype with multiple organ system involvement and signina limb reductions.

Minor corrections:

Page 5 lines 132-133: "In addition, few microdeletions at 19p13.11-p12 have been identified in individuals with neurodevelopmental delay..." should read: "In addition, a few microdeletions at 19p13.11-p12 have been identified in individuals with neurodevelopmental delay..."

Page 7 lines 177-178: "...collectively referred to as CdLS epesignature." should read: "collectively referred to as the CdLS epesignature."

Version 1:

Reviewer comments:

Reviewer #1

(Remarks to the Author)

I thank the authors for their responses to my prior comments (Reviewer 1) and feel that the issues raised have been addressed. I appreciate their additional work on the entire manuscript.

A few minor comments:

- Please use growth "restriction" instead of "retardation" in the Introduction.
- In general, it is not recommended to use the gene name before "individual" (or "patient"). Thus, an "individual with a MAU2 variant" is preferable to a "MAU2 individual". This is likely up to the editors.

Reviewer #2

(Remarks to the Author)

In this revised manuscript by Parenti and colleagues, the authors have addressed all my immediate concerns and, as far as I can tell, those of the two other reviewers. In particular, I appreciate the detailed and clear explanations about cohort delineation and description, the reformulation of the two-hybrid assay result presentation and interpretation together with careful association with their respective clinical features that have been added to the results and discussion sections. Altogether, it strengthens the manuscript and its main message – MAU2 variants as a novel, distinct CdLS subtype. Having read the other reviewers' comments and this revised ms, I agree that this discovery does indeed deserve publication, and that it will be of interest for Nature Communications readership.

Reviewer #3

(Remarks to the Author)

REVIEWER COMMENTS

We thank the reviewers for their constructive and insightful comments, which have helped us to improve the clarity and quality of our manuscript. A point-by-point response to each concern is provided below, with reviewer comments shown in black and our responses in blue.

Reviewer #1 (Remarks to the Author):

This work by Parenti et al. is a very important well-written article describing individuals with MAU2 variants, previously minimally reported and likely with many variants lethal. They make convincing arguments that MAU2 be considered to be a new CdLS-causal gene in that certain variants found in MAU2 cause CdLS as supported both clinically and by epigenetic signature; this is also supported by findings in the MAU2 knockout mouse. The detailed mouse description is excellent, and brain findings very interesting and novel. The mammalian two-hybrid results also are supportive of MAU2's complementary role with NIPBL. MAU2 being a CdLS-causal gene has long been suspected, so this article will be well received.

The patients with the variants are thoroughly described, including using the international consensus clinical diagnostic criteria for CdLS. It is somewhat less clearly stated that those patients with lower clinical diagnostic criteria scores or less typical findings as seen in Table 1 do not resemble patients with CdLS, including in Figure 6. (It is also unfortunate that most of the patients' photos are not available for the manuscript.)

We have now explicitly clarified in the Results section that individuals with scores ≥ 9 exhibited facial or clinical features that overlap with the CdLS spectrum, whereas those with scores < 9 did not show any such overlapping features (lines 361-363).

We agree with this Reviewer that the small number of available photographs limits the visual documentation of the full cohort. Unfortunately, despite repeated and extensive efforts, the majority of families did not consent to sharing images. However, to strengthen the link between clinical appearance and CdLS scoring, we have now added the CdLS diagnostic scores to the legend of Figure 6 (now Figure 7) for individuals with available photographs, thus enabling a more immediate connection between facial dysmorphism and CdLS overlap.

It would be helpful if the authors could describe a non-CdLS MAU2 facial phenotype or if they feel that it is nonspecific.

In light of the comments of Reviewer 1 and Reviewer 2, we have substantially expanded the comparative clinical analysis within the Results section (lines 337-353) and added a new Table (Table 2). The manuscript now includes:

1. **A direct comparison between MAU2 cases with molecularly confirmed CdLS and those without molecular evidence of CdLS.**

We now explicitly report the frequencies of canonical CdLS features in MAU2-CdLS versus non-CdLS MAU2 individuals (Table 2).

2. **A comparison between MAU2-CdLS and classic NIPBL-CdLS.**

We added a detailed analysis comparing MAU2-CdLS cases with individuals carrying

pathogenic *NIPBL* variants. We now describe in the manuscript which CdLS traits overlap (e.g., eyebrow features) and which are attenuated in *MAU2*-CdLS (e.g. mouth involvement, limb findings, developmental severity).

3. A clear statement regarding the recognizability of *MAU2*-CdLS.

We now emphasize in the manuscript that there are no unique or immediately distinguishable dysmorphic features that allow a clinician to identify a patient specifically as having *MAU2*-CdLS, rather than other subtypes of CdLS.

In addition, patient 5 (non CdLS) appears to have a repaired cleft lip, but only cleft palate is mentioned in the supplemental table. Clefting is not mentioned otherwise in Table 1, and typically only cleft palate is seen in patients with CdLS. I wondered if any of the other patients (especially those without the CdLS higher clinical scores or CdLS epismature) had cleft lip, and if so, could that be associated with *MAU2* (or if this was the only one and thus likely unrelated)?

We thank the reviewer for carefully noting this discrepancy. We confirm that cleft lip had not been explicitly reported in the original version of Supplementary Table 3. This has now been corrected by adding a dedicated row ("Cleft lip and/or palate") in Supplementary Table 3 (currently Supplementary Table 5), and the presence of bilateral cleft lip and palate is now explicitly indicated for Individual 5.

It is now visible in the updated Supplementary Table 3 (now Supplementary Table 5) that no other patient in the cohort displays cleft lip or cleft palate. Therefore, at present, clefting represents a single-occurrence feature (1/18) in our series and cannot be considered a recurrent or distinguishing phenotype associated with *MAU2* variants. For this reason, clefting was not included in Table 1, which is intended to summarize the most frequent clinical features observed in the cohort.

Importantly, Individual 5 carries a variant that does not impair the interaction with *NIPBL* and displays a very low CdLS clinical score. Although epismature analysis could not be performed due to lack of material, both the clinical assessment and the mammalian two-hybrid results support classification of this variant within the non-CdLS *MAU2* category. While we discuss in the manuscript the possibility that this variant may act through an alternative pathogenic mechanism related to the role of *MAU2* in transcriptional regulation rather than as *NIPBL*-chaperone, the available data do not allow us to draw definitive conclusions regarding clefting as part of this mechanism.

It would be helpful to mention that hearing loss, seen so commonly in CdLS, was not seen in most patients with the *MAU2* variants, since that may set it apart.

The low frequency of hearing impairment in our *MAU2* cohort has now been specified in the Results paragraph "Variants in *MAU2* result in variable phenotypes mainly characterized by short stature and microcephaly", where *MAU2*-CdLS is compared to canonical CdLS (lines 349-351). Overall, the lower frequency of hearing impairment in *MAU2*-CdLS aligns with the generally attenuated CdLS phenotype observed in these patients in comparison with *NIPBL*-CdLS.

Finally, in the abstract and elsewhere, microcephaly and short stature are described as being consistent features, but really are not present in 20-30% of the patients according to Table 1; perhaps the word "consistent" should be replaced with "major" or "in the majority" or something less universal?

The manuscript has been revised accordingly, and microcephaly and short stature are now described as “the two major clinical features associated with MAU2 variants” to reflect their high, but not universal, prevalence.

Otherwise, no further comments, and strongly recommend publication.

We thank the reviewer for their positive assessment and recommendation for publication

Reviewer #2 (Remarks to the Author):

Cornelia de Lange syndrome (CdLS) is a neurodevelopmental disorder of genetic origin that is characterised by typical features including craniofacial dysmorphism, hirsutism, growth and cognitive delays, as well as system impairment such as gastrointestinal tractus and heart. Of note, clinical manifestations are of variable severity amongst patient population, and while typical, severe CdLS cases are easily diagnosed, diagnosis for the lower end of the spectrum often needs to discriminate between CdLS and other related neurodevelopmental syndromes. In that aim, an international effort has established score-based guidelines to provide clinicians with common diagnosis practices.

CdLS arises from heterozygous pathogenic variants in a number of genes encoding members of cohesin complex and of its regulators. Vast majority of cases (up to 70 %) involve NIPBL gene, which encodes the protein NIPBL that, together with its binding partner MAU2, forms the Kollerin complex and loads cohesin complexes onto chromatin. Importantly, Kollerin also acts in the regulation of gene expression independently of cohesin. 15 % of remaining cases are accounted for by variants in cohesin-related genes (SMC3, SMC1A, HDAC8, RAD21 and BRD4) and are characterised by broad clinical manifestations and, in some cases, atypical presentations. So far, only two patients with variants in the NIPBL partner MAU2 have been reported, including one by the authors.

Pathophysiological mechanisms for MAU2 variants are lacking due to the small number of cases, albeit impaired NIPBL/MAU2 interaction and protein folding and stability have been proposed by the same authors. In addition, and for the same reason, it is unclear whether MAU2 variants result in classical CdLS presentation or constitute a distinct subtype of CdLS. In this new manuscript entitled “Pathogenic variants in the cohesin loader subunit MAU2 lead to a new Cornelia de Lange Syndrome subtype”, Parenti and colleagues report eighteen patients carrying different types of variants in MAU2, which they used to interrogate common and distinct clinical features and disease-relevant molecular mechanism.

Once having presented the 15 distinct MAU2 variants they identified and that span its entire coding sequence, the authors first assessed patient DNA methylation profiles (episignature) aiming at defining or refining CdLS episignatures. Out of 14 cases studied, 10 of them showed a Episign methylation profile similar to that previously established for CdLS patients. As four cases did not match CdLS episignature, the authors tested the possibility of specific MAU2 episignature(s) and were able to establish two distinct MAU2 Episignatures, which both allow clustering of most MAU2 patients, though with some discordances.

As an altered physical interaction between NIPBL and MAU2 impairs NIPBL folding and stability, and therefore its transcriptional activity, the authors next addressed the impact of the ten MAU2 substitution variants in mammalian two-hybrid assays. These experiments revealed that 6 out of the 10 tested variants did impact negatively on the ability of NIPBL/MAU2 to activate transcription of reporter gene (luciferase). In addition, the authors assessed to which extent MAU2 variants impacted on MAU2 stability as a cause for reduced luciferase expression by probing MAU2 protein levels produced in these assays by immunoblots, which did not allow to address this particular point unambiguously.

Next, the authors assessed the consequence of truncating, out-of-frame variants on the amounts of NIPBL and MAU2 at protein and mRNA levels using patient-derived dermal fibroblasts obtained from one patient and her father from whom variant was inherited. RT-qPCR experiments revealed that MAU2 mRNA levels are reduced as compared to control individuals while the situation for NIPBL is unclear. Immunoblotting analysis of MAU2 and NIPBL in the same cell samples suggested to some extent a reduction of both protein levels in the patient's cells and her father's.

With a unique cohort of 18 "MAU2 patients" at their disposal, the authors then exhaustively analysed their clinical features in order to identify subsets of them that would show some specificity/peculiarity between the whole MAU2 cohort compared to the 11 MAU2 cases molecularly assigned to CdLS. This comparison highlighted a slightly higher frequency in the CdLS MAU2 patients as compared to the whole cohort of a few facial features while frequency in all other features were similar between the two groups of MAU2 patients.

Finally, the authors present the characterisation of a *Mau2* heterozygous null mouse, which exhibits shorter size, microcephaly and a set of structural brain anomalies, in agreement with previously reported CdLS mouse models, and recapitulates some of CdLS hallmark features.

Overall, this manuscript is an interesting piece as it reports on the largest MAU2 patient cohort published so far, which represents useful resources for clinicians, and unambiguously establishes MAU2 variants as a cause of CdLS. Further, the authors report that MAU2 patients present specific DNA-methylation based epigenatures, which are distinct from that of the established CdLS epigenature and may well have utility in clinical practices. Parenti and colleagues also evidenced how these variants impact on the ability of the Kollerin complex to form, possibly hinting at pathophysiological mechanisms.

This study largely focuses on the presentation and phenotypical characterisation of the MAU2 cohort, and, albeit mammalian two-hybrid assay does point toward impaired interaction between MAU2 and NIPBL, do not improve significantly molecular understanding of CdLS. Along the same line, the presented mouse model constitutes an interesting confirmation of MAU2 haploinsufficiency as a cause of murine CdLS-like features, but does bring any information as to how it results into described brain structure anomalies. Furthermore, and if most of this manuscript reads well, some parts are a bit confusing, both concerning results and text (see below for specific comments).

In conclusion, I would recommend the authors to consider publishing their work in a journal with a scope more focused in human genetics, although I would be happy to reconsider my position if the authors were able to improve significantly on the molecular mechanism at play here.

We thank the reviewer for the thoughtful evaluation of our work. We would like to highlight that the primary aim of this study is to establish *MAU2* as a new disease-causing gene. To this end, we combine human genetic data, experimental data, and an *in vivo* mouse model that recapitulates key phenotypes. Together, these complementary approaches extend beyond phenotypic description and provide convergent evidence supporting the pathogenicity of *MAU2* variants.

We fully agree that further studies will be essential to delineate the downstream molecular mechanisms linking *MAU2* haploinsufficiency to specific neurodevelopmental abnormalities. While an in-depth mechanistic dissection is beyond the scope of the present manuscript, our study already provides important new insights into *MAU2* function and establishes a solid foundation for future investigations. Notably, the molecular assays developed and rigorously validated here offer a robust functional framework for the interpretation of emerging *MAU2* variants, thereby directly facilitating variant classification and informing clinical decision-making. We anticipate that these tools will enable subsequent studies to further unravel the biological roles of *MAU2* and refine our understanding of the associated disease mechanisms. In this context, we believe that our work is well aligned with the

scope of *Nature Communications*, as it reports the identification and comprehensive validation of a previously unrecognized disease gene with clear translational relevance.

Main comments (in no particular order):

1- Regarding description of Figure 2, the authors wrote “ three of these variants..., but grouped with controls when analysed separately. ” (l. 194-197). This is a bit unclear to me what it means and what the implications of such a result are (Sup Fig 1). This will need more detailed description.

We agree that our earlier formulation was unclear. We have now expanded on how evaluation of the variants was performed (lines 187-195 in the manuscript).

Basically, in the full cohort, 10 of 14 *MAU2* samples were positive for the CdLS epesignature and therefore showed strong similarity to CdLS reference cases. Because *MAU2* samples also share *MAU2*-specific methylation features, all *MAU2* cases, including those negative for the CdLS epesignature, tended to cluster together as a group, positioning them on the CdLS side of the heatmap. When the CdLS-positive *MAU2* samples were removed, this cohort effect was eliminated, allowing the four CdLS-negative *MAU2* samples to be evaluated independently. Under these conditions, their methylation profiles showed greater similarity to controls than to CdLS reference cases, consistent with their negative EpiSign classification.

2- Similarly confusing to me is the identification of the two *MAU2* epesignatures. Were they obtained from distinct clustering analyses? Were only 9 and 7 samples considered for the signature discovery? If so, why? Which *MAU2* sample falls into each signature? What about non *MAU2* CdLS cases (i.e., are *MAU2* signatures discriminating between *MAU2* and other CdLS?

To clarify how the *MAU2*-specific epesignatures were identified, we have included a more thorough description in the manuscript (lines 199-215).

In addition, we have now added two new Supplementary Tables (Supplementary Tables 3 and 4) containing the lists of selected differentially methylated probes of each *MAU2* epesignature and further comment on the different *MAU2*-specific epesignatures at lines 223-230.

3- Regarding Figure 3 (currently Figure 4): I agree with the authors that while six variants impact on detected luciferase activity, four do not; however I am not convinced by the corresponding interpretation that *MAU2* epesignature has much to do with the results: indeed, all tested variants that show reduced luciferase activity exhibit CdLS signature, regardless of *MAU2* signatures. Similarly, all variants that do not affect luciferase are negative for the CdLS signature, irrespective of the *MAU2* signature. Therefore, and unless I missed something, this assay almost perfectly identifies CdLS signature positive and negative cases, while it does not do so for *MAU2* ones.

We agree that the mammalian two-hybrid luciferase assay primarily discriminates variants according to CdLS epesignature status, rather than *MAU2* epesignature status, and that this distinction was not sufficiently explicit in the original text.

Our intention was to highlight that variants disrupting the *MAU2*–NIPBL interaction consistently associate with a positive CdLS epesignature, supporting a close link between impaired *MAU2*–NIPBL complex formation and the classical CdLS molecular phenotype. We agree, however, that the assay does not provide direct evidence for *MAU2*-specific molecular effects independent of NIPBL, and we

have therefore revised the manuscript to avoid any assumptions that cannot be substantiated by the data (lines 252-258).

The manuscript has been revised to explicitly state that the luciferase assay correlates with CdLS episignature status rather than *MAU2* episignature status. We have therefore removed assumptions not directly supported by the assay and now present isolated *MAU2* episignatures without attributing them to specific molecular mechanisms. All interpretations are now aligned with the experimental data.

In the same set of experiments, presented in Sup Fig 4 (currently Sup Fig 2), results of protein expression analyses are not very clear: NIPBL and MAU2 seem to be challenging to blot for and, as a consequence, it is very hard to assess clearly differences in protein amounts of MAU2, NFKB and GAPDH on the presented images. Same comment applies to Flag immunoblot. Finally, the fact that the authors argue that differences in protein amount caused by different variants depend at least partly on the plasmid used for ectopic expression further confuses interpretation for the reader.

We agree that the effects of the variants on protein stability were not properly addressed in the original version of the manuscript. To clarify this point, we have removed the FLAG immunoblot from the Supplementary Figure (currently Supplementary Figure 2), because it was indeed confusing and was not adding relevant information. To address the stability issue more directly, we have now performed cycloheximide (CHX) chase assays, and the results of these new experiments are now outlined in lines 266-272. All but one variants exhibited little to no change in stability relative to the wild-type protein, although some showed statistically significant differences with small effect size. In contrast, the p.(Ala309_Lys322del) variant displayed a pronounced reduction in stability. These findings indicate that the pathogenicity of the p.(Ala309_Lys322del) variant likely arises from a combination of impaired NIPBL interaction and decreased protein stability, a factor that should be considered when interpreting the mammalian two-hybrid assay results.

4- In Figure 4 (currently Figure 5): I acknowledge authors' efforts and will of completeness as they assessed mRNA abundance expressed relative to two different normalisation genes. However, if MAU2 mRNA reduction is observed in cells from the patient and from her father independently of the normalisation gene used, the fact that it shows a reduction for NIPBL in one case and no changes in the other one is surprising, and adds difficulty to interpretate these results.

We agree that the apparent discrepancy for *NIPBL* between normalization strategies complicates interpretation. To strengthen our conclusions, we included an additional housekeeping gene, *SNAPIN*, and re-analyzed the existing RNA samples. One control sample could not be included in the new analysis due to insufficient remaining material; for consistency, this sample was also excluded from all normalizations (*NADH* and *GAPDH*), resulting in a final control group of four individuals for all experiments.

Inspection of control distributions across experiments (see Reviewers-comments_Figure1) showed that *GAPDH* exhibited the highest variability across control samples, indicating that it is a less reliable reference in this system. In contrast, *NADH* and *SNAPIN* displayed more stable expression. On this basis, we now focus on *NADH* and *SNAPIN* as reference genes in the manuscript and have removed *GAPDH*-normalized data.

Reviewers-comments_Figure1. Distributions of controls across housekeeping genes

During the revision of our manuscript, we also recognized that our initial statistical approach, which treated all six measurements per individual (two technical replicates for each of three RNA preparations) as independent, could overestimate significance. In line with current qPCR best practices, we now average technical replicates within each RNA preparation and perform statistical analyses on the resulting three biological replicates per individual. With this approach, *MAU2* mRNA levels remain consistently reduced in both the patient and her father across reference genes, supporting a robust biological effect (Figure 5). For *NIPBL*, no reduction is now observed in either the patient or the father when normalized to *NADH* or *SNAPIN* (Figure 5). Given the instability of *GAPDH* and the lack of concordance across stable reference genes or between the two mutation carriers, we interpret the previous *NIPBL* decrease observed with *GAPDH* as a normalization artifact rather than a true biological effect.

The limited number of biological replicates ($n = 3$ per individual) restricts statistical power, which likely explains why the reduction of *MAU2* in the father, when normalized to *SNAPIN*, does not reach statistical significance. An additional RNA preparation from new fibroblast cultures would have further strengthened the analysis; however, this is currently not feasible due to poor cell growth of the two fibroblasts lines carrying the *MAU2* frameshift variant. Overall, the revised results using validated reference genes and appropriate statistical analysis support the conclusion that *MAU2* downregulation is reproducible and biologically meaningful, whereas the apparent *NIPBL* reduction with *GAPDH* normalization is unlikely to reflect a genuine effect.

5- In Figure 5 (currently Figure 6). The comparison of canonical clinical features between whole MAU2 cohort vs MAU2 patients that exhibit CdLS signature highlights some differences at the facial level, which is interesting per se. How different would it be if CdLS positive patients vs CdLS negative patients were compared? It might reveal clear differences between CdLS and MAU2. Also, how does the 18 patient MAU2 cohort compare to classic CdLS patients (e.g., carrying *NIPBL* variants)? Any significant differences may strengthen the reality of a MAU2-specific subgroup of CdLS.

In light of the comments of Reviewer 1 and Reviewer 2, we have substantially expanded the comparative clinical analysis within the Results section (lines 337-353) and added a new Table (Table 2). The manuscript now includes:

1. A direct comparison between MAU2 cases with molecularly confirmed CdLS and those without molecular evidence of CdLS.

We now explicitly report the frequencies of canonical CdLS features in MAU2-CdLS versus non-CdLS MAU2 individuals (Table 2).

2. A comparison between MAU2-CdLS and classic NIPBL-CdLS.

We added a detailed analysis comparing MAU2-CdLS cases with individuals carrying pathogenic NIPBL variants. We now describe in the manuscript which CdLS traits overlap (e.g., eyebrow features) and which are attenuated in MAU2-CdLS (e.g., mouth involvement, limb findings, developmental severity).

3. A clear statement regarding the recognizability of MAU2-CdLS.

We now emphasize in the manuscript that there are no unique or immediately distinguishable dysmorphic features that allow a clinician to identify a patient specifically as having MAU2-CdLS, rather than other subtypes of CdLS.

Minor comments

1- In Figure 1, degrees of variation intolerance, as stated in the legend, are represented by different colours. I guess these degrees are also represented by heights of the curves at the different positions. If so; please indicate in the legend, and if not, please precise.

We have now added the explanation for the intolerance in the legend of Figure 1

2- As identification of MAU2 epesignatures represent a main result of the presented work, I feel like Sup Fig 2 and 3 would nicely fit in the main text (in Figure 2 or as a distinct figure).

The panels related to the MAU2-specific epesignature are now included in the manuscript as a main Figure (Figure 3).

3- I would replace Fig4E with SUP Fig6, as it gives much clearer indication of MAU2 protein reduction. Ideally, NIPBL would be blotted on the same experiment.

We agree that the MAU2 signal shown in Supplementary Fig. 6 (now Supplementary Figure 4) is of higher quality. However, Fig. 4E and Supplementary Fig. 6 were generated independently in two different laboratories and with different objectives. Specifically, the blot shown in Fig. 4E (now Figure 5E) was performed in-house and was designed to assess MAU2 and NIPBL levels on the same membrane, whereas Supplementary Fig. 6 (now Supplementary Figure 4) was generated by a collaborator and was specifically optimized for the detection of MAU2.

We would indeed have preferred to obtain a NIPBL blot of comparable quality under the same experimental conditions used for Supplementary Fig. 6 (now Supplementary Figure 4). Unfortunately, despite extensive optimization efforts, probing for NIPBL under these conditions was unsuccessful. This difficulty is consistent with previous reports, as NIPBL is known to be challenging to detect by western blotting.

For this reason, we chose to retain Fig. 4E (now Figure 5E) in the main figure, as it allows the simultaneous visualization of both MAU2 and NIPBL protein levels from the same experiment, which we believe is important for the interpretation of the data. Supplementary Fig. 6 (now Supplementary Figure 4) is therefore provided to complement Fig. 4E (now Figure 5E) by illustrating the MAU2 reduction with higher signal quality.

Reviewer #3 (Remarks to the Author):

Variants in *Mau2* causing a CdLS-like phenotype have been described previously (as acknowledged by the authors) – this manuscript describes the largest cohort to date and characterizes the effects these variants mechanistically and in an animal model – adding credence and support to the pathogenicity of these variants and expanding the cohesin-related genes that contribute to the causality of CdLS and related diagnoses. The manuscript is well written and structured with synergistic clinical, molecular, cellular and animal modeling to support the authors' hypotheses and claims. Parenti et al., propose (and provide convincing data) that pathogenic variants in MAU2 result in a CdLS subtype and that MAU2 should be recognized as a new CdLS-associated gene. This work will contribute significantly to the field by expanding our understanding of key regulators of cohesin's role in contributing to mammalian structural and developmental differences and expanding our understanding how these proteins regulate developmental gene expression.

1) Recommend replacing the terms “disease” and “disorder” with “diagnosis”

We thank the reviewer for this thoughtful suggestion. After careful consideration, we feel that in several instances the terms “disease” and “disorder” remain the most precise descriptors of the pathological condition itself, whereas “diagnosis” refers to the process or outcome of clinical classification. For example, Cornelia de Lange syndrome is appropriately described as a neurodevelopmental disorder rather than a “neurodevelopmental diagnosis.” For this reason, we have retained the original terminology where it is scientifically most accurate.

2) Recommend replacing “patients” with “individuals” when describing affected people

In agreement with the reviewer's suggestion, the term “patients” has been replaced with “individuals” throughout the manuscript.

3) How do the authors explain that one of their pathogenic variants was found in gnomAD (p.(Cys50Ser)) in 6 individuals? This variant in the reported individual was paternally inherited – what was the father's phenotype (in the supplementary table they list him as 165 cm tall – which plots onto the standard growth curves)? The methylation pattern for this individual also clusters with the controls and the two-hybrid assay suggests benign.

The individual carrying the p.(Cys50Ser) variant (Individual 6) presents with an extremely mild phenotype, limited to short stature and very non-specific dysmorphic features (long and smooth philtrum, thick lower vermilion). We therefore consider it plausible that individuals carrying this variant in the general population may remain clinically unrecognized, which could explain the presence of the variant in gnomAD.

For this variant, the methylation profile clusters with controls rather than with CdLS individuals, and the mammalian two-hybrid assay does not demonstrate a clear functional effect. However,

p.(Cys50Ser) is nonetheless associated with the *MAU2*-specific episignature, suggesting milder molecular consequences that may not reach the threshold required to generate a classical CdLS episignature. This is consistent with the attenuated clinical presentation observed in Individual 6. Taken together, these findings support the conclusion that p.(Cys50Ser) is associated with milder functional and clinical consequences, which may explain both its presence in population databases and the very mild phenotype of Individual 6. Importantly, under ACMG criteria, this variant still remains a variant of uncertain significance (VUS), in contrast to the majority of other variants described in this study, which are classified as likely pathogenic or pathogenic. We have now explicitly incorporated into the Discussion (lines 455-460) the possibility that variants exerting only modest functional effects may give rise to correspondingly mild—and potentially clinically unrecognized—phenotypes.

As highlighted by this reviewer, the variant was paternally inherited. The only available clinical information for the father is his height (165 cm), which corresponds to -1.6 SD based on [editorial note: confidential information redacted] corresponding population growth charts and therefore does not meet criteria for short stature (the standard deviation for the father's height has now been added to Supplementary Table 5). Following the reviewer's comment, the corresponding physician attempted to re-contact the father to obtain additional clinical information, but this was not possible, and no further phenotypic data are available. In light of this premises (variant with weak functional consequences and very mild phenotype), it becomes inherently difficult to distinguish incomplete penetrance from complete penetrance with extremely mild expressivity. In other words, the available data do not allow us to determine whether in this case the father is entirely unaffected or whether he may display very subtle or subclinical features. By contrast, for the other inherited variants in our cohort that are associated with more pronounced molecular effects and clearer clinical phenotypes, the carrier parents consistently display at least some clinical features overlapping with those of their affected children. This pattern supports the interpretation that penetrance is generally high for variants with strong functional and phenotypic consequences.

Nevertheless, because detailed phenotypic information is not uniformly available for all carrier parents, we cannot formally exclude incomplete penetrance in individual cases. We have therefore revised the manuscript to explicitly acknowledge this limitation and to adopt appropriately cautious wording regarding penetrance (rows 499-502).

4) Would be ideal to include all parents who carry the variants argued to be causative in the "affected" individuals in this manuscript, although the authors state that they were unable to obtain sufficient clinical information on all of these individuals – unclear if all parents who carry these variants are truly "affected" but more information would help delineate the clinical spectrum of *MAU2*-related CdLS, and whether or not non-penetrance is possible (or if some of these variants may in fact be benign). Possible to include any photographs of affected/carrier parents?

We agree with the reviewer that including carrier parents would be useful to delineate the clinical spectrum of *MAU2*-related CdLS.

Carrier parents of individuals 11 and 16 are already included in the cohort; however, both declined consent for publication of photographs. The carrier parents of individual 6 and of the sisters 14/15 were not included due to insufficient clinical information.

We made repeated efforts to obtain additional clinical details and photographs from these two parents, but these attempts were unsuccessful. Consequently, the available data are insufficient for

their inclusion in the cohort. As detailed phenotypic information is not uniformly available for all carrier parents, incomplete penetrance cannot be formally excluded in individual cases. We have revised the manuscript to explicitly acknowledge this limitation and have adopted appropriately cautious wording regarding penetrance (rows 499-502).

5) Patient 5 – looks like also may have cleft lip in photo (listed as “midline cleft palate”)?

We thank the reviewer for carefully noting this discrepancy. We confirm that cleft lip had not been explicitly reported in the original version of Supplementary Table 3 (Now Supplementary Table 5). This has been corrected by adding a dedicated row (“Cleft lip and/or palate”) in Supplementary Table 3 (currently Supplementary Table 5), and the presence of bilateral cleft lip and palate is now explicitly indicated for Individual 5.

6) Do the authors know if all individuals treated with GH were actually GH deficient on testing (they mention it for a few but for others – presume that information was not available) – if GH levels were not known in these individuals it should be stated in Suppl Table 3.

We agree that information on GH–IGF-1 axis evaluation is highly important for the clinical management. To address this point, we have contacted all the corresponding physicians to verify whether formal GH–IGF-1 axis testing had been performed. We have now added a dedicated “GH deficiency” row to Supplementary Table 5.

Based on the additional information obtained, 7 individuals had undergone GH–IGF-1 axis evaluation, 3 of whom were confirmed to have GH deficiency. For 11 individuals, data regarding GH testing were not available (NA). These details are now incorporated into Supplementary Table 5 and referenced in the revised text (lines 463-464).

7) Can the authors expand on how this cohort was identified – the bias may be that they were enrolled in various CdLS studies (“international collaborators”) and/or selected from individuals obtaining exome or genome sequencing (e.g. through GeneMatcher ascertainment) which would aggregate them with likely syndromic or non-syndromic developmental delay of intellectual disability that might introduce a bias in the clustering of phenotypic features around the various testing parameters undertaken as part of this study.

We have now clarified the referral reasons for genetic analysis in the Methods section (lines 559–562). Only four individuals were referred directly to us by collaborators because of our CdLS expertise; all other cases were identified through GeneMatcher based on the presence of *MAU2* variants (genotype-driven instead of phenotype-driven). Referral indications for the diagnostic analysis were heterogeneous (see also “Clinical Diagnosis” row in Supplementary Table 5) and included CdLS (2x), suspected cohesinopathy (2x), syndromic microphthalmia (1x), developmental delay (2x), syndromic intellectual disability (1x), short stature (5x), microcephalic primordial dwarfism (1x), short stature and microcephaly (2x), recurrent pneumothoraces (1x). This diversity reduces the likelihood that the cohort might be heavily biased toward any single phenotypic feature.

8) It would be of interest for the authors to expand a bit (beyond the one sentence mention of the double heterozygous mouse model) as to why there is significant phenotypic discrepancy between NIPBL related CdLS and MAU2-related CdLS? Given the direct interaction and functional facilitation of these two proteins one might expect a more significant overlap between the phenotypes – especially

in those with haploinsufficient mutational mechanisms – but only severe pathogenic variants in CdLS cause the truly “classical” severe CdLS phenotype with multiple organ system involvement and signina limb reductions.

We agree that the relatively mild average phenotype observed in individuals with *MAU2* variants, particularly truncating alleles, appears counterintuitive given the strong mutual dependence of *MAU2* and *NIPBL* for protein stability and the marked dosage sensitivity of *NIPBL*. This has also been a central question for us, and we now address it explicitly in the revised Discussion (lines 480-497).

In this study, we show that *MAU2* variants either impair the *MAU2*–*NIPBL* interaction (in-frame variants) or lead to secondary reduction of *NIPBL* protein levels (truncating variants). In addition, we have previously shown (Parenti et al., 2020) that the transcriptome of fibroblasts from an individual with an in-frame *MAU2* deletion shows a strong positive correlation with that from a fibroblast cell line carrying a *NIPBL* nonsense variant ($R = 0.79$). This correlation supports the notion that *MAU2* and *NIPBL* perturb a largely shared regulatory program. At the same time, the correlation is not complete, implying that a subset of genes is differentially affected. Moreover, transcriptomic similarity in a single cell type does not necessarily equate to equivalent functional dosage across all tissues, developmental stages, or regulatory thresholds *in vivo*, all of which are critical determinants of CdLS severity.

We therefore propose that the aforementioned phenotypic differences between *MAU2*- and *NIPBL*-related CdLS may arise from a combination of quantitative and subunit-specific effects. First, the reduction of *NIPBL* secondary to *MAU2* haploinsufficiency may be less pronounced than primary *NIPBL* haploinsufficiency resulting from a direct *NIPBL* mutation. Previous studies have shown that *NIPBL* transcript and protein levels vary substantially among affected individuals and that lower *NIPBL* expression correlates with more severe clinical outcomes. Thus, even if downstream transcriptional signatures in fibroblasts are strongly aligned, the effective reduction of *NIPBL* function *in vivo* may be less pronounced, less uniform across tissues, or less likely to cross critical developmental thresholds in typical *MAU2* haploinsufficiency than in classical *NIPBL* loss-of-function.

Second, there is precedent for gene-specific expressivity within the cohesin pathway itself. Pathogenic variants in *SMC1A*, *SMC3*, and *RAD21* all target the same cohesin ring, yet they are associated with recognizably different clinical phenotypes. This is generally attributed to subunit-specific roles (e.g. distinct contributions to DNA binding, ring architecture or regulatory interfaces) and allele-dependent quantitative effects, rather than uniform disruption of a single, identical function. By analogy, *MAU2* and *NIPBL*, although forming an obligate cohesin loader and being mutually stabilizing, are not functionally interchangeable. Structural and functional work suggests that *NIPBL* provides the main DNA-binding and cohesin-engaging surfaces, whereas *MAU2* contributes to N-terminal shielding of *NIPBL*, complex assembly and is known to have a role in transcriptional regulation. Perturbations in each subunit are therefore expected to yield overlapping but non-identical disease manifestations, just as *SMC1A*, *SMC3*, and *RAD21* variants do within the cohesin ring.

Finally, tissue-specific sensitivity to cohesin loader dosage, together with genetic background, likely determines whether molecular perturbations produce the classical, multisystem CdLS phenotype.

Minor corrections:

Page 5 lines 132-133: “In addition, few microdeletions at 19p13.11-p12 have been identified in individuals with neurodevelopmental delay...” should read: “In addition, a few microdeletions at 19p13.11-p12 have been identified in individuals with neurodevelopmental delay...”

The manuscript has been adapted accordingly

Page 7 lines 177-178: "...collectively referred to as CdLS epesignature." should read: "collectively referred to as the CdLS epesignature."

The manuscript has been adapted accordingly